# An ESCRT-LEM protein surveillance system is poised to directly monitor the nuclear envelope and nuclear transport system

David J Thaller[1], Matteo Allegretti[2], Sapan Borah[1], Paolo Ronchi[3], Martin Beck[2], C Patrick Lusk[1]*

[1]Department of Cell Biology, Yale School of Medicine, New Haven, United States; [2]Structural and Computational Biology Unit, European Molecular Biology Laboratory, Meyerhofstrasse, Germany; [3]Electron Microscopy Core Facility, European Molecular Biology Laboratory, Meyerhofstrasse, Germany

**Abstract** The integrity of the nuclear membranes coupled to the selective barrier of nuclear pore complexes (NPCs) are essential for the segregation of nucleoplasm and cytoplasm. Mechanical membrane disruption or perturbation to NPC assembly triggers an ESCRT-dependent surveillance system that seals nuclear pores: how these pores are sensed and sealed is ill defined. Using a budding yeast model, we show that the ESCRT Chm7 and the integral inner nuclear membrane (INM) protein Heh1 are spatially segregated by nuclear transport, with Chm7 being actively exported by Xpo1/Crm1. Thus, the exposure of the INM triggers surveillance with Heh1 locally activating Chm7. Sites of Chm7 hyperactivation show fenestrated sheets at the INM and potential membrane delivery at sites of nuclear envelope herniation. Our data suggest that perturbation to the nuclear envelope barrier would lead to local nuclear membrane remodeling to promote membrane sealing. Our findings have implications for disease mechanisms linked to NPC assembly and nuclear envelope integrity.
DOI: https://doi.org/10.7554/eLife.45284.001

*For correspondence:
patrick.lusk@yale.edu

## Introduction

The molecular machinery that biochemically segregates the nucleus and the cytoplasm has been extensively investigated. The foundational components of this selective barrier include the double-membrane nuclear envelope with embedded nuclear pore complexes (NPCs). NPCs impose a 'soft' diffusion barrier to macromolecules larger than ~40 kD (*Popken et al., 2015*; *Timney et al., 2016*) while providing binding sites for the rapid and selective transport of signal-bearing (nuclear localization and nuclear export signals; NLSs and NESs) macromolecules, which are ferried through the NPC by shuttling nuclear transport receptors (NTRs; a.k.a. karyopherins/importins/exportins; *Schmidt and Görlich, 2016*). Directionality and energy for NTR-selective transport is imparted by the spatial segregation of the Ran-GTPase whose nuclear GTP-bound form destabilizes and stabilizes import and export complexes, respectively (*Floch et al., 2014*).

Interestingly, the robustness of the nuclear envelope barrier has been shown to be compromised in several different contexts, including in diverse human diseases (*Hatch and Hetzer, 2014*; *Lusk and King, 2017*). For example, there is an emerging body of work linking the function of NTRs and NPCs with neurodegenerative diseases like ALS and FTD (*Nousiainen et al., 2008*; *Freibaum et al., 2015*; *Jovičić et al., 2015*; *Kaneb et al., 2015*; *Zhang et al., 2015*; *Kim and Taylor, 2017*; *Shi et al., 2017*). These studies, coupled to the observations of age-related declines in

**eLife digest** With the exception of bacteria, living cells contain most of their DNA inside a structure called the nucleus. The membranes of the nucleus form a protective wall around the DNA, while pores within this wall act as entry check-points, controlling what can and cannot get inside. Maintaining the structure of this wall is critical for cell survival. Problems can occur if the nuclear wall or its pores become disrupted, as in the case of cancer and neurodegenerative diseases. Thankfully cells have developed a protective surveillance system that can rapidly identify and repair any damage made to the nuclear wall. However, how this damage is found and what activates its repair is poorly understood.

Now, Thaller et al. have investigated two key proteins that they suspected were involved in the surveillance of the nuclear border in budding yeast: Chm7 and Heh1. Chm7 is part of a complex group of proteins that can cut and sculpt the shape of membranes, while Heh1 is normally embedded on the inside of the nucleus. Thaller et al. discovered that, when the nuclear wall is disrupted, Heh1 recruits Chm7 to the site of damage and activates it. Once activated Chm7 can repair the damage to the nuclear wall, by sealing over defective nuclear pores and closing gaps caused by breakages.

Thaller et al. showed that the transport system that normally moves molecules into and out of the nucleus also imports Heh1 and actively excludes Chm7, physically segregating them to opposite sides of the nuclear border. If the nuclear wall becomes damaged this leads to the local meeting of Heh1 and Chm7 at these sites. Heh1 will then activate the membrane shaping mechanisms of Chm7, rapidly repairing the nuclear border in response to the damage.

It is possible cell structures other than the nucleus use a similar surveillance system to protect their borders. Manipulating the border surveillance system of the nucleus could be used to treat the detrimental impacts of damage caused to the nuclear wall by disease.

DOI: https://doi.org/10.7554/eLife.45284.002

NPC function in both post-mitotic multicellular systems (*D'Angelo et al., 2009*; *Savas et al., 2012*; *Toyama et al., 2013*) and also in replicative aging models like budding yeast (*Janssens et al., 2015*; *Lord et al., 2015*), support a theme in which the function of the nuclear envelope could be mitigatory of age-related disease progression (*Schreiber and Kennedy, 2013*; *Jevtić et al., 2014*; *Serebryannyy and Misteli, 2018*). Of similar interest, the hallmark cellular pathophysiology of early-onset dystonia are nuclear envelope herniations (*Goodchild et al., 2005*) that emanate from NPC-like structures (*Laudermilch et al., 2016*). As analogous herniations have been observed in many genetic backgrounds associated with defects in NPC biogenesis in yeast over several decades (*Thaller and Lusk, 2018*), this has contributed to the idea that the herniations might be the result of either defective NPC assembly events (*Scarcelli et al., 2007*; *Onischenko et al., 2017*; *Zhang et al., 2018*) and/or the triggering of a NPC (*Wente and Blobel, 1993*) or NPC assembly quality control pathway (*Webster et al., 2014*; *Webster et al., 2016*). The latter could depend on the function of the endosomal sorting complexes required for transport (ESCRT), a membrane scission machinery that has been proposed to seal-off malforming NPCs (*Webster et al., 2016*).

That there could be mechanisms to surveil the assembly of NPCs makes considerable sense as there are hundreds of NPCs, each containing hundreds of nucleoporins/nups (*Kosinski et al., 2016*; *Kim et al., 2018*), that are assembled during interphase in mammalian cells (*Maul et al., 1972*; *Doucet et al., 2010*; *Dultz and Ellenberg, 2010*). There are approximately 100 NPCs formed during a budding yeast cell cycle, which includes a closed mitosis (*Winey et al., 1997*). As interphase NPC assembly likely occurs through an inside-out evagination of the inner nuclear membrane (INM) followed by membrane fusion with the outer nuclear membrane (ONM) (*Otsuka et al., 2016*), holes are constantly being formed in the nuclear envelope. Without mechanisms to surveil this process, de novo NPC biogenesis might pose a threat to nuclear-cytoplasmic compartmentalization (*Webster et al., 2014*). Consistent with this idea, malformed or damaged NPCs are not passed on to daughter cells in budding yeast (*Colombi et al., 2013*; *Makio et al., 2013*; *Webster et al., 2014*). Further, deletion of the ESCRT machinery in the context of genetic backgrounds where nuclear envelope herniations have been observed for example *nup116Δ* (*Wente and Blobel, 1993*)

or *apq12Δ* (*Scarcelli et al., 2007*) cells require a nuclear envelope-specific ESCRT, Chm7 (the ortho-logue of mammalian CHMP7), for viability (*Bauer et al., 2015*; *Webster et al., 2016*). While we have previously proposed that a biochemical signature of malforming NPCs is surveilled by integral inner nuclear membrane proteins of the Lap2-emerin-MAN1 (LEM) domain family, specifically Heh2, it remains to be formally established what the signal that leads to ESCRT recruitment to the nuclear envelope actually comprises (*Webster et al., 2014*).

Evidence that the ESCRT machinery acts at holes in the nuclear envelope is further exemplified by their critical role in performing annular fusion events during the final stages of nuclear envelope ref-ormation at the end of mitosis in mammalian cells (*Olmos et al., 2015*; *Olmos et al., 2016*; *Vietri et al., 2015*; *Gu et al., 2017*; *Ventimiglia et al., 2018*). Moreover, ESCRTs are also required for the efficient repair of nuclear ruptures that arise during the migration of cells through tight con-strictions (*Denais et al., 2016*; *Raab et al., 2016*). And, it is most likely that they also act to repair nuclear envelope ruptures that are induced by intracellular mechanical stresses from either the actin cytoskeleton (*Hatch and Hetzer, 2016*; *Robijns et al., 2016*), or from those observed during telo-mere crisis (*Maciejowski et al., 2015*). Lastly, recent work also suggests a role for ESCRTs in the context of turning over NPCs in quiescent cells (*Toyama et al., 2019*). It remains an open question, however, whether the mechanisms that repair nuclear ruptures, seal the nuclear envelope at the end of mitosis, and protect against defective NPC assembly respond to an identical upstream signal and proceed through the same membrane-sealing mechanism.

Clues to what might constitute the upstream signal that leads to nuclear envelope-recruitment of ESCRTs could be drawn from other contexts where ESCRTs protect membrane compartments including endolysosomes (*Radulovic et al., 2018*; *Skowyra et al., 2018*) and the plasma membrane (*Jimenez et al., 2014*; *Scheffer et al., 2014*; *Gong et al., 2017*). In both of these cases, there is evi-dence to suggest that the local release of $Ca^{2+}$ is a trigger for ESCRT recruitment, through (at least at the plasma membrane) a $Ca^{2+}$ binding protein, ALG-2 (*Jimenez et al., 2014*; *Gong et al., 2017*). Whether $Ca^{2+}$ plays a role at the nuclear envelope remains unaddressed. More generally, there are two, often redundant, recruitment mechanisms seeded by either an ESCRT-I, II complex and/or ESCRT-II and ALIX (Bro1 in yeast) that bind and activate ESCRT-III subunit polymerization (*Wemmer et al., 2011*; *Henne et al., 2012*; *Tang et al., 2015*; *Tang et al., 2016*; *Christ et al., 2016*) on specific membranes throughout the cell (reviewed in *Schöneberg et al., 2017*; *McCullough et al., 2018*).

ESCRT-III polymers predominantly made up of the most abundant ESCRT-III (Snf7/CHMP4B) scaf-fold negative but also in at least one case, positive membrane curvature (*McCullough et al., 2015*), and directly contribute to membrane scission (*Adell et al., 2014*; *Adell et al., 2017*; *Schöneberg et al., 2018*). The AAA +ATPase Vps4 disassembles ESCRT-III filaments by directly interacting with MIM (MIT interacting motif) domains present on a subset of ESCRT-III subunits (*Obita et al., 2007*; *Stuchell-Brereton et al., 2007*; *Agromayor et al., 2009*; *Xiao et al., 2009*; *Han et al., 2015*) by threading the ESCRT-III filaments through the central cavity of a hexameric ring (*Yang et al., 2015*; *Han et al., 2017*; *Monroe et al., 2017*; *Su et al., 2017*). It is likely that ESCRT-III disassembly by Vps4 directly contributes force to promote membrane scission (*Schöneberg et al., 2018*). Whether the membrane scission reaction is different in distinct subcellular contexts like at the nuclear envelope remains to be understood.

Consistent with the idea that there might be unique ESCRT membrane remodeling mechanisms at play in distinct compartments, a step-wise recruitment and activation mechanism requiring the ESCRT-II Vps25 and the ESCRT-III Vps20 is thought to be required at budding yeast endosomes (*Saksena et al., 2009*; *Teis et al., 2010*; *Tang et al., 2015*; *Tang et al., 2016*), but both of these proteins are absent from the genetic and biochemical analyses of the nuclear envelope arm of the ESCRT pathway (*Webster et al., 2014*; *Webster et al., 2016*). These data suggest that other pro-teins likely contribute to ESCRT-III activation at the nuclear envelope. Key candidates are Chm7 and the inner nuclear membrane (INM) proteins, Heh1/Src1 (orthologue of LEM2) and Heh2 (orthologue of MAN1 or other LEM-domain proteins). These proteins have collectively been shown to interact biochemically and genetically with Snf7 (*Webster et al., 2014*; *Webster et al., 2016*) and Heh1 is required for the focal accumulation of Chm7 at the nuclear envelope in genetic backgrounds where NPC assembly is inhibited (*Webster et al., 2016*). Remarkably, the interactions between Heh1 and Chm7 are well conserved in both fission yeast (*Gu et al., 2017*) but also in mammalian cells, where LEM2 is required to recruit CHMP7 to the reforming nuclear envelope at the end of mitosis

(**Gu et al., 2017**). It remains to be understood, however, whether LEM proteins (or Chm7) directly contribute to ESCRT-III activation at the nuclear envelope or whether additional proteins are involved.

Heh1 and Heh2 contain an N-terminal helix-extension-helix (heh) motif (the LEM domain), followed by an INM targeting sequence that (at least in the case of Heh2 but likely also Heh1 [**King et al., 2006**; **Lokareddy et al., 2015**]) includes an NLS and a ~ 200 amino acid unstructured region that are both required for INM targeting (**Meinema et al., 2011**). They both also contain a second nuclear-oriented domain, which likely folds into a winged helix (WH) (also called MAN1/Src1-C-terminal homology domain or MSC; **Caputo et al., 2006**); this domain is also well conserved through evolution (**Mans et al., 2004**; **Mekhail et al., 2008**). The LEM domain proteins as a family have been ascribed diverse roles in gene expression either through binding to transcription factors, BAF, or the lamins (**Barton et al., 2015**). While yeasts lack BAF and lamins, the LEM domain proteins nonetheless directly interface with chromatin (**Grund et al., 2008**; **Barton et al., 2015**), contribute to rDNA repeat stability (**Mekhail et al., 2008**) and the mechanical robustness of the nucleus (**Schreiner et al., 2015**). This latter function is likely revealed by the observation in many diverse yeast species that loss of Heh1 leads to nuclear envelope disruption (**Yewdell et al., 2011**; **Gonzalez et al., 2012**; **Yam et al., 2013**), however, it might also reflect Heh1's role in recruiting ESCRTs to the nuclear envelope. Thus, the relationship between how the LEM proteins contribute to nuclear integrity and ESCRT-mediated surveillance remains to be clearly defined.

In the following, we further explore the molecular determinants of Chm7 recruitment to the budding yeast nuclear envelope by Heh1. We determine that the spatial segregation of Chm7 and Heh1 on either side of the nuclear envelope is driven by NTRs and the robustness of the nuclear transport system. Perturbations of this system, or the exposure of Heh1 to the cytosol leads to the local recruitment and Heh1 WH-dependent activation of Chm7. At sites of Chm7 hyperactivation, we observe remarkable alterations to nuclear envelope morphology including nuclear envelope herniations and intranuclear INM invaginations suggesting a role for membrane expansion and remodeling during nuclear envelope repair.

## Results

### Chm7 is actively exported from the nucleus by Xpo1

It was our previous experience that visualizing endogenous levels of Chm7-GFP was challenging due to its low level of expression (**Webster et al., 2016**), thus, to gain further insight into the localization determinants of Chm7 in budding yeast, we overexpressed Chm7-GFP behind the control of a galactose (*GAL1*) inducible promoter. As shown in **Figure 1C**, culturing of cells in galactose for a short time (~45 min) led to the appearance of Chm7-GFP fluorescence throughout the cytosol. Unexpectedly, we also observed that Chm7-GFP was excluded from the nuclear interior (orange asterisks). These data raised the possibility that Chm7-GFP is unable to cross the diffusion barrier imposed by NPCs, or, there is an active nuclear export pathway that prevents Chm7 from accessing the nucleus.

Consistent with the possibility that Chm7 might be recognized by an export NTR, we identified two putative leucine-rich NESs at the C-terminus of Chm7 using the NES prediction algorithm, LocNES (**Xu et al., 2015**; **Figure 1A,B**). Interestingly, the higher scoring predicted NES (NES2) overlapped with the potential MIM1 motif (**Bauer et al., 2015**) (**Figure 1A**, **Figure 1—figure supplement 1A**). Indeed, the Chm7 putative MIM1 motif stands out from that of other budding yeast ESCRT-IIIs because of an additional isoleucine that contributes the fourth hydrophobic amino acid required for an effective class 1a NES (**Figure 1—figure supplement 1A**). A second leucine in the middle of this region also aligns with the predictive spacing of residues in class 1b NESs (**Figure 1—figure supplement 1A**). Moreover, these putative NESs were conserved in Chm7 orthologues in other species of yeast, mice, flies and humans, although are curiously absent from *C. elegans* (**Figure 1—figure supplement 1B**). As LocNES predicts NESs specific for the major export NTR, Xpo1/Crm1, we tested whether the inhibition of Xpo1 reversed the nuclear exclusion of Chm7-GFP. For these experiments, we took advantage of the *xpo1-T539C* allele, which sensitizes budding yeast Xpo1 to the Xpo1 inhibitor Leptomycin B (LMB) (**Neville and Rosbash, 1999**) (**Figure 1E**). As shown in **Figure 1C,** 45 min LMB treatment of cells expressing Chm7-GFP led to the striking accumulation

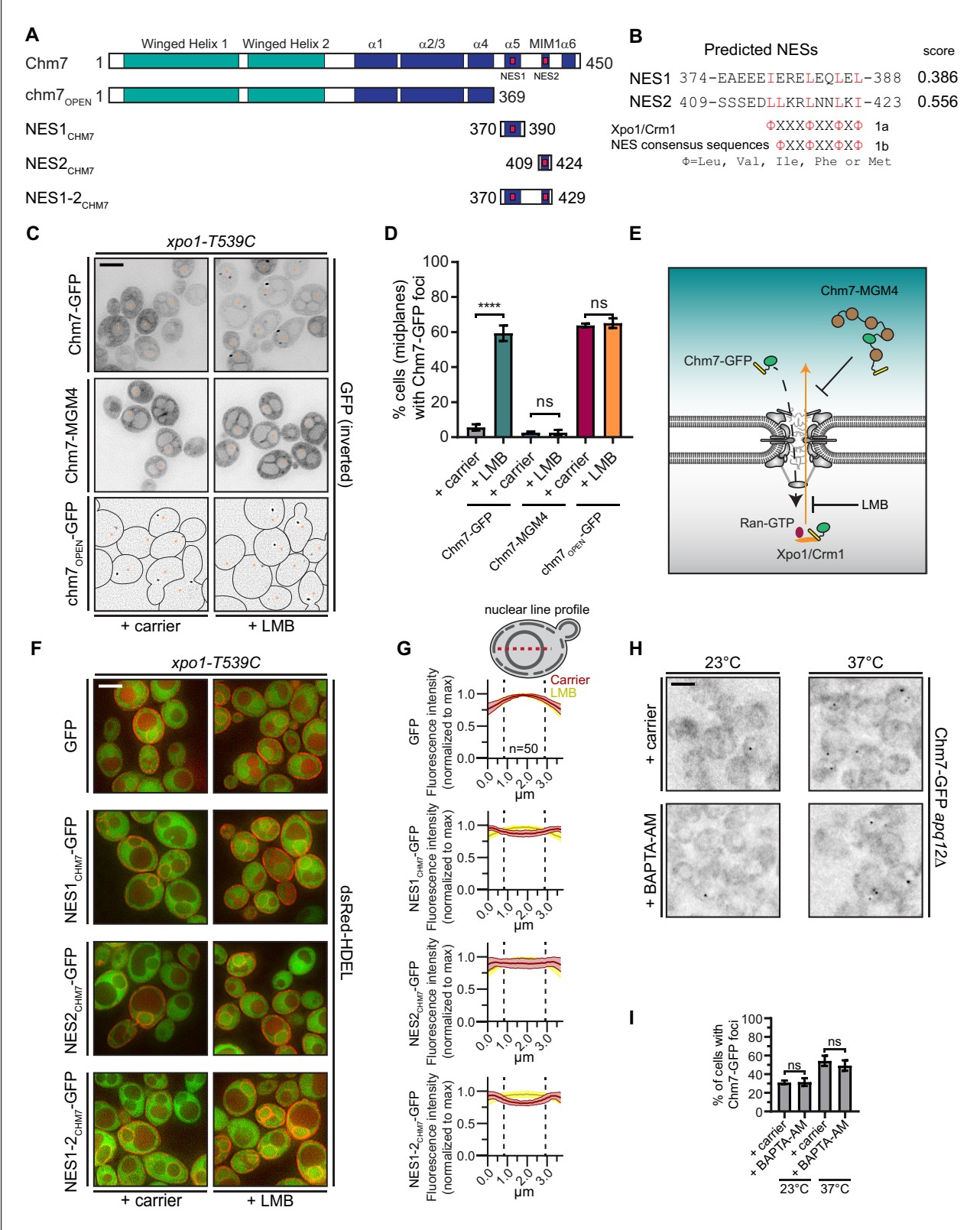

**Figure 1.** Chm7 can diffuse across the NPC but is actively exported by Xpo1. (A) Schematic of Chm7 and deletion constructs with predicted winged helix domains (teal), alpha helices (blue) and NESs (red boxes); numbers are amino acids. (B) Predicted Chm7 NESs with probability score from LocNES; numbers are amino acids from Chm7 sequence. Hydrophobic residues in putative NESs are highlighted red as per the consensus class 1a and 1b NESs shown. (C) Deconvolved inverted fluorescence micrographs of the indicated Chm7-GFP constructs in a LMB-sensitive strain (*xpo1-T539C*) treated with

*Figure 1 continued on next page*

*Figure 1 continued*
carrier (MeOH) or LMB. Nuclei are marked with orange asterisks. Because of the lack of detectable cytosolic fluorescence in chm7$_{OPEN}$-GFP-expressing cells, cell boundaries in the bottom panel are drawn from phase-images (not shown). Scale bar is 5 μm. (D) Plots showing the percentage of cells with Chm7-GFP nuclear envelope foci from C. Data are from three independent replicates where > 100 cells were counted for each strain. Only images of midplanes were quantified. *P*-values from unpaired Student's t-test where ns is p>0.05, ****p≤0.0001. (E) Schematic of the experiment and interpretation of C. (F) Deconvolved fluorescence micrographs (merge of green and red channels) of the indicated GFP, and GFP-NES constructs co-expressed with dsRed-HDEL to help visualize the nucleus. Cells were treated with carrier (MeOH) or LMB before imaging. Scale bar is 5 μm. (G) To quantify relative nuclear exclusion of the GFP and GFP-NES constructs in F, line profiles bisecting the nucleus as shown in diagram were measured from 50 cells/condition pooled from three independent replicates. The normalized average (thick lines) ±SD (thin lines) of the carrier (red) and LMB-treated (yellow) cells are shown. Vertical dotted lines designate nuclear boundaries. (H) Ca$^{2+}$ chelation does not affect Chm7-GFP recruitment to the nuclear envelope in *apq12Δ* cells. Deconvolved inverted fluorescence micrographs of Chm7-GFP in *apq12Δ* cells pretreated for 30 min with carrier (DMSO) or BAPTA-AM and shifted to the indicated temperature for 45 min. (I) Plot quantifying the percentage of *apq12Δ* cells with Chm7-GFP foci at the indicated temperature and treatment. three independent replicates of >100 cells were quantified per replicate. *P*-values are calculated from un-paired Student's t-test where ns is p>0.05.
DOI: https://doi.org/10.7554/eLife.45284.003
The following figure supplement is available for figure 1:

**Figure supplement 1.** Chm7 can diffuse across the NPC but is actively exported by Xpo1.
DOI: https://doi.org/10.7554/eLife.45284.004

of Chm7-GFP at the periphery of the nucleus most often in highly fluorescent foci in over 60% of cells (*Figure 1D*), which is an underestimate as only midplanes were quantified.

The focal accumulation of Chm7-GFP at the nuclear periphery suggested that Chm7 was able to enter the nucleus upon Xpo1 inhibition. Therefore, to distinguish whether nuclear entry was driven by an active NTR-mediated nuclear import pathway, or, whether it was the result of passive diffusion across the NPC, we generated a Chm7-GFP fused to five maltose-binding proteins (Chm7-MGM4); this ~280 kD protein would be extremely inefficient at transiting through the NPC unless it contained an NLS. Consistent with the idea that Chm7-GFP's entry into the nucleus was governed by diffusion and not active NLS-mediated transport, the distribution of Chm7-MGM4 was indistinguishable from Chm7-GFP but, in contrast, incubation with LMB had no effect on its localization (*Figure 1C–E*).

As inhibition of Xpo1 led to Chm7-GFP accumulation at the nuclear envelope, we reasoned that it was likely that the prediction of NESs in Chm7 was likely accurate. However, as both NLS and NES prediction is of limited utility, we directly tested whether the predicted NESs were indeed sufficient to prevent an inert GFP reporter from localizing in the nucleus at steady state. We tested the localization of GFP-fusions to each predicted Chm7 NES alone, and in combination (*Figure 1A*). To help demark the nuclear boundary, these constructs were expressed alongside dsRED-HDEL, which localizes throughout the continuous nuclear envelope/ER lumen (*Madrid et al., 2006*). As shown in *Figure 1F*, when compared to GFP-alone, all three NES$_{CHM7}$-GFP constructs showed a deficit of nuclear accumulation that could be reversed by treatment with LMB. Interestingly, when we compared line profiles of GFP fluorescence drawn from the cytosol and bisecting the nucleus, NES1$_{CHM7}$-GFP was more obviously excluded from the nucleus than NES2$_{CHM7}$-GFP with NES1-2$_{CHM7}$-GFP showing the most striking absence of nuclear signal (compare the depth of the 'valleys' of the red lines; *Figure 1G*). Therefore, it is likely that both predicted NESs contribute to the efficient export of Chm7. Consistent with this, the examination of a truncation of Chm7 (chm7$_{OPEN}$; *Figure 1A*) lacking both NESs dramatically accumulates in one or two foci on the nuclear envelope in a way that is not impacted by LMB (*Figure 1C,D* and see *Webster et al., 2016*). Thus, the steady state nuclear exclusion of Chm7 in wildtype cells is determined by its passive diffusion into the nucleus and the Xpo1-mediated recognition of NESs in Chm7. That, having entered the nucleus, Chm7 accumulates in a focus along the nuclear periphery is consistent with its binding and activation at the INM. The latter being reflected in its focal accumulation, which would be consistent with a polymerization event.

That Chm7 can be recruited and activated at the INM without any perturbation to the nuclear envelope raises the possibility that there are no other upstream signals that are necessary to trigger Chm7 recruitment. While this is difficult to conclusively prove, we nonetheless tested whether Ca$^{2+}$ could reflect an additional signal because of its role in other ESCRT-mediated membrane repair processes (*Jimenez et al., 2014*; *Scheffer et al., 2014*; *Gong et al., 2017*; *Skowyra et al., 2018*). As

Chm7 is recruited to the nuclear envelope in *apq12Δ* cells when grown at elevated (37°C) temperatures (*Webster et al., 2016*), we evaluated whether this recruitment was influenced by chelating $Ca^{2+}$ using BAPTA-AM. As shown in *Figure 1H*, there was no obvious change to the number of Chm7-GFP foci that appear during the temperature shift in the presence or absence of $Ca^{2+}$ (*Figure 1H,I*). Thus, it is unlikely that a $Ca^{2+}$ signal is a major contributor to this pathway.

## Cytosolic exposure of the Heh1 WH domain is sufficient to recruit Chm7 to membranes

We hypothesized that Chm7 was excluded from the nucleus in order to prevent its untimely or inappropriate 'activation' in the absence of a perturbation of the nuclear envelope barrier. Such a model predicts that there must be a nuclear-binding partner that itself might be 'hidden' from cytosolic Chm7; based on our and others' prior work (*Webster et al., 2014*; *Webster et al., 2016*; *Gu et al., 2017*) the most obvious candidate was Heh1. To test this hypothesis, we generated deletion constructs of Heh1 coupled to the Red Fluorescent Protein (RFP) expressed behind the *GAL1* promoter (note that there is vacuolar autofluorescence even under repressed glucose conditions, see asterisks in *Figure 2A*). Unlike many other INM proteins that tend to back up into the ER upon overexpression (*Lusk et al., 2007*), Heh1-RFP continues to accumulate at the INM even at high levels due to its use of an active NTR-dependent INM targeting pathway (See *Figure 2A,B* and *King et al., 2006*). Thus, even when overexpressed at levels that we estimate to be an order of magnitude higher than endogenous levels, the majority of Heh1 is localized to the INM and would be predicted to be inaccessible to cytosolic Chm7 (*Figure 2A*, galactose, middle panels). Consistent with this, we observed no change to the steady state distribution of endogenously-expressed Chm7-GFP, which includes a minor fraction within a nuclear envelope focus in ~30% of cells (*Webster et al., 2016*; *Figure 2A*). Deletion of the LEM domain of Heh1 also had no effect on Chm7-GFP distribution as heh1(51-834)-RFP was also exclusively localized at the INM (*Figure 2A*).

We next tested deletions that encompassed the putative NLSs in Heh1 including heh1(303-834), and heh1(442-834), which resulted in the accumulation of these truncations throughout the cortical ER. Strikingly, we observed a concurrent re-distribution of Chm7-GFP into foci that colocalized with the RFP signal (*Figure 2A*). In the case of heh1(442-834), only the WH domain is available for Chm7 binding. Consistent with this, there was a complete lack of Chm7-GFP at the ER in cells expressing heh1(442-735), where the WH is removed (*Figure 2A*). Thus, exposure of the Heh1 WH domain to the cytosol is both necessary and sufficient to recruit Chm7-GFP to ER membranes.

We next assessed the functional importance of the Heh1-WH domain to *apq12Δ* cells, which require both *CHM7* and *HEH1* for full viability (*Yewdell et al., 2011*; *Bauer et al., 2015*; *Webster et al., 2016*) (*Figure 2C*). Interestingly, the loss of fitness observed in *heh1Δapq12Δ* cells could only be rescued by the gene encoding full length Heh1 or the *heh1(51-834)* allele. In contrast, deletions that resulted in Heh1 mistargeting or those that are unable to recruit Chm7 (e.g. *heh1(1-735)*, which lacks the coding sequence for the WH domain) were unable to fully complement growth. Thus, while the WH domain is important, the N-terminal INM targeting domain is also a critical component of Heh1 functionality in the context of *apq12Δ* cells.

## Chm7 binds to an INM platform

The localization data clearly pointed to a direct interaction between the WH domain of Heh1 and Chm7. Unfortunately, we were unable to detect a stable interaction in vitro with purified recombinant proteins (one example shown in *Figure 3—figure supplement 1A*), although we note such an interaction has been shown with the human versions of these proteins (*Gu et al., 2017*). While there are many potential reasons for these negative data, one possibility is that there are additional proteins (or lipids) that contribute to the interaction in vivo. To test this idea, we affinity purified Chm7-GFP from whole cell extracts using anti-GFP nanobody-coupled beads (*Figure 3—figure supplement 1B*) and subjected protein eluates to MS/MS peptide identification. Consistent with the idea that Chm7 is localized throughout the cytosol in a potentially inactive form, we detected few specific peptide spectra with the exception of Chm7 itself when compared to proteins derived from wildtype cell extracts that bind non-specifically to the anti-GFP beads (*Figure 3A*). To facilitate visualization, we directly relate the average spectral counts (two experiments) from bound fractions of affinity purifications of Chm7-GFP and no-GFP controls in *Figure 3A*.

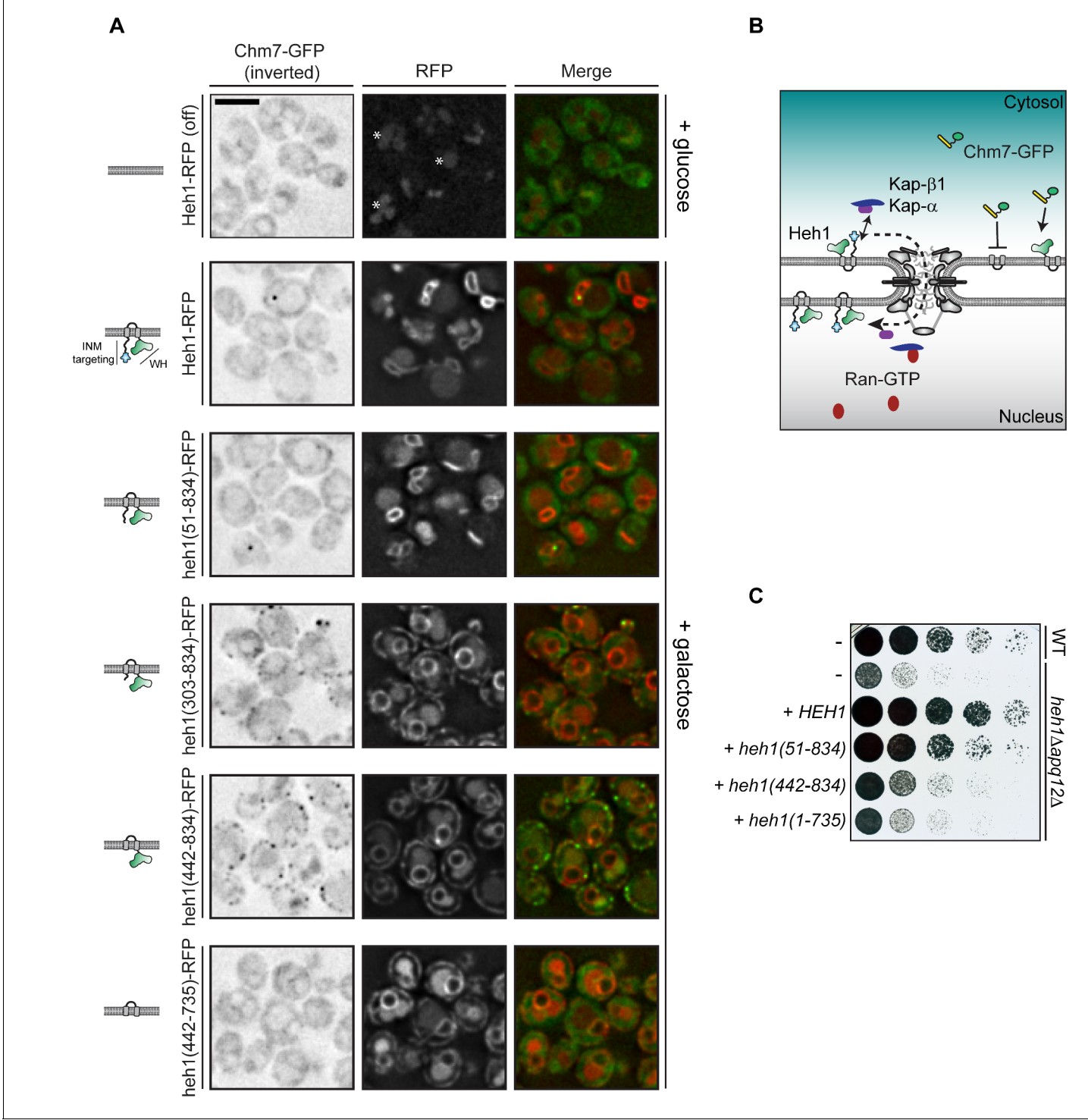

**Figure 2.** Cytosolic exposure of the Heh1 WH domain is sufficient to recruit Chm7 to ER membranes. (A) Deconvolved fluorescence micrographs of Chm7-GFP (inverted) either prior to (+glucose) or after 2 hr of overexpression (+galactose) of RFP-tagged full length and truncations of Heh1 (depicted in cartoons at left in a lipid bilayer). Asterisks indicate vacuolar autofluorescence in the red channel. Scale bar is 5 μm. (B) Cartoon of experiment and interpretation of A. The efficient INM targeting of Heh1 depends on Kap-α/Kap-β1 (blue and purple) and on Ran-GTP (red). (C) Tenfold serial dilutions of the indicated strains spotted onto YPG plates to express the indicated truncations of Heh1. Plates imaged after growth at 30°C for 36 hr.
DOI: https://doi.org/10.7554/eLife.45284.005

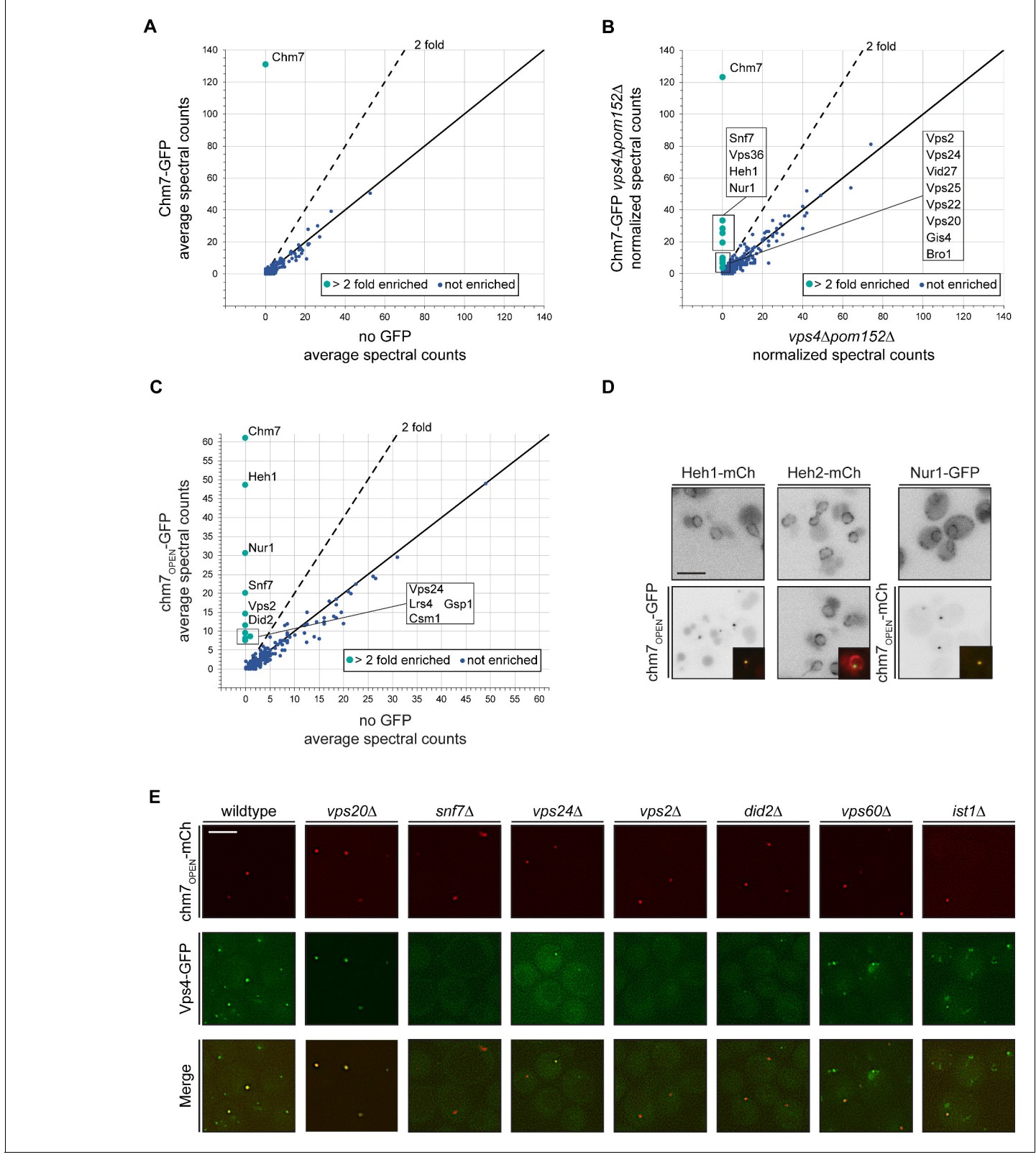

**Figure 3.** Chm7 binds to Heh1, Nur1 and downstream ESCRTs required for Vps4 recruitment. (**A–C**) Affinity purifications of Chm7-GFP and chm7$_{OPEN}$-GFP were performed from wildtype and *vps4Δpom152Δ* cells. Bound proteins were eluted and subjected to LC-MS/MS peptide identification. Scatter plots of the number of peptide spectra identifying the indicated proteins were directly compared to those identified in 'no GFP' samples and were generated from data shown in *Figure 3—source data 1*. Dotted line and teal circles represent peptides found at least two fold enriched over non-

*Figure 3 continued on next page*

*Figure 3 continued*

specific proteins (blue). Plots A and C represent an average of two replicates of normalized spectral counts of peptides identified by MS. (D) Heh1 and Nur1 but not Heh2 colocalize with chm7$_{OPEN}$. Deconvolved inverted fluorescence micrographs of Heh1-mCherry (red channel), Heh2-mCherry (red channel) and Nur1-GFP (green channel) are shown in indicated columns in both the top and bottom panels. Either chm7$_{OPEN}$-GFP or chm7$_{OPEN}$-mCherry are co-expressed as indicated in bottom panels but chm7open localization only shown in inset as merge of green and red channels. (E) Vps4 recruitment to chm7$_{OPEN}$ requires Snf7 and downstream ESCRTs. Deconvolved fluorescence micrographs of Vps4-GFP and chm7$_{OPEN}$-mCherry in the indicated strain backgrounds. Green, red and merged images shown.

DOI: https://doi.org/10.7554/eLife.45284.006

The following source data and figure supplement are available for figure 3:

**Source data 1.** Table containing all identified MS peptide spectra from affinity purifications of Chm7-GFP and chm7$_{OPEN}$-GFP in either WT or *vps4Δpom152Δ* cells with no-GFP controls.
DOI: https://doi.org/10.7554/eLife.45284.008
**Figure supplement 1.** Chm7 binds to Heh1, Nur1 and downstream ESCRTs required for Vps4 recruitment.
DOI: https://doi.org/10.7554/eLife.45284.007

We therefore turned to examining the interactome of Chm7-GFP under conditions in which it accumulates at the nuclear envelope, for example in *vps4Δpom152Δ* cells, which we had previously shown leads to Chm7-GFP accumulation within a nuclear envelope domain enriched for malformed NPCs (*Webster et al., 2014*; *Webster et al., 2016*). Shot-gun MS identification of peptides derived from bound proteins to Chm7-GFP now revealed specific interactions with several ESCRTs including Snf7 and Vps36 (*Figure 3B*). Most interestingly, dozens of spectra specific for Heh1 were identified. Considering Heh1 is a low abundant integral membrane protein (measured to be as low as 428 molecules/cell; *Kulak et al., 2014*), this result was particularly striking. In addition, another low abundant (~354 molecules/cell; *Kulak et al., 2014*) integral INM protein, Nur1 was also detected. As Nur1 is known to interact with Heh1 within the CLIP (chromosome linkage INM proteins) complex (*Mekhail et al., 2008*), these data suggest that Chm7 engages Heh1 within a broader INM platform, at least in the context of cells lacking *VPS4*. Of note, no components of the NPC were specifically detected, nor was Heh2.

We next tested binding partners of chm7$_{OPEN}$-GFP (*Figure 3—figure supplement 1B*), which also provides a potential mimic of the physiological circumstances when Chm7 is recruited to the nuclear envelope. In this case, Heh1 was the top hit (*Figure 3C*). In addition to Nur1, other members of CLIP were also specifically identified including Lrs4 and Csm1. Curiously, Gsp1 (budding yeast Ran) was also found (*Figure 3C*). Further, alongside Snf7, other ESCRT-IIIs including Vps2, Vps24 and Did2 were detected (*Figure 3C*). In contrast to bound proteins purified with Chm7-GFP in the *vps4Δpom152Δ* cells, we did not detect any specific peptides for ESCRT-II subunits. We surmise this is likely because Chm7-GFP can be seen in cytosolic foci in *vps4Δ* cells (See *Webster et al., 2016* and Figure 5A), whereas chm7$_{OPEN}$-GFP exclusively localizes to the nuclear envelope. As a further test of the specificity of the interactions between chm7$_{OPEN}$-GFP and integral INM proteins, we observed the near-quantitative accumulation of both Heh1 and Nur1 fluorescent fusion proteins (produced at endogenous levels) at the chm7$_{OPEN}$ focus, while the distribution of Heh2 was unaltered (*Figure 3D*).

We noted that although we detected several additional ESCRT-III proteins in the affinity purifications of chm7$_{OPEN}$-GFP, we did not detect any peptides for Vps4. Thus, we investigated whether a functional Vps4-GFP fusion (*Adell et al., 2017*) could also be specifically recruited to the chm7$_{OPEN}$ focus at the nuclear envelope, which it was in virtually all cells (*Figure 3E*). The fluorescence intensity of Vps4-GFP could be correlated to that of the chm7$_{OPEN}$-mCherry focus, suggesting a close relationship between the number of Chm7 and Vps4 molecules recruited to this nuclear envelope site (*Figure 3—figure supplement 1C*).

We therefore next assessed the molecular determinants of Vps4-GFP recruitment to the chm7$_{OPEN}$ site by first focusing on deleting the genes encoding ESCRT-III subunits found in the chm7$_{OPEN}$-GFP affinity purifications including *SNF7*, *VPS24*, *VPS2*, and *DID2* (*Figure 3C*). In all these deletion strains, Vps4-GFP recruitment to chm7$_{OPEN}$-mCherry foci was reduced or, in the case of *snf7Δ* cells completely eliminated, with minimal impact on the accumulation of chm7$_{OPEN}$-mCherry itself (*Figure 3E*; *Figure 3—figure supplement 1D,E*); although we noted that the average area encompassed by the chm7$_{OPEN}$-mcherry focus was increased in *snf7Δ* cells (*Figure 3E*; *Figure 3—*

*figure supplement 1F*). These data support a model in which Vps4 may be recruited to the nuclear envelope using a similar cohort of ESCRT-III interactions as those observed at endosomes (*Babst et al., 2002*). Consistent with this idea, we also tested *ist1Δ* and *vps60Δ* cells, which encode proteins that impact Vps4 recruitment (*Dimaano et al., 2008*; *Rue et al., 2008*) and ATPase activation (*Azmi et al., 2008*; *Yang et al., 2012*) at endosomes. In both of these strains, there was a modest decrease of Vps4-GFP fluorescence at the chm7$_{OPEN}$-mCherry foci (*Figure 3E*; *Figure 3—figure supplement 1D*).

In marked contrast to the disruption of ESCRT-III components that act downstream of Snf7, deletion of *VPS20*, which acts upstream of Snf7 at endosomes (*Teis et al., 2008*) and is absent from the nuclear envelope (*Webster et al., 2014*; *Webster et al., 2016*), we observed a remarkable ~3 fold increase of Vps4-GFP at the nuclear chm7$_{OPEN}$ focus (*Figure 3E*, *Figure 3—figure supplement 1D*). These data suggest that in the absence of the endosome ESCRT arm, there is a larger pool of Vps4 that is available to interact with the chm7$_{OPEN}$ network at the nuclear envelope. Importantly, the total Vps4-GFP protein levels were not notably altered in any of the ESCRT deletion backgrounds tested (*Figure 3—figure supplement 1G*). Together, these data reinforce that there are unique components of the endosome and nuclear envelope ESCRT pathways, but Vps4 can nonetheless be recruited to the INM alongside or downstream of Snf7.

## Fluorescence ESCRT Targeting and Activation (FETA) Assay

Interestingly, as shown in *Figure 3D*, chm7$_{OPEN}$ is able to shift the distribution of both Heh1 and Nur1 from an evenly-distributed nuclear peripheral localization to one that is co-localized with the chm7$_{OPEN}$ focus. This raised the formal possibility that Chm7 recruitment to the nuclear envelope might in fact be independent of and precede binding to Heh1; in such a model Heh1 would be required for its focal accumulation, which we interpret to be 'activation'. Thus, this result illustrated the need to develop a better controlled experimental system where the mechanism of Chm7 recruitment and activation can be decoupled. We thus conceived of an experimental approach that we termed the F̲luorescent E̲SCRT T̲argeting and A̲ctivation Assay (FETA; *Figure 4A*). FETA exerts both temporal control over the expression of Chm7-GFP (through the *GAL1* promoter) in addition to spatial control over its recruitment to the nuclear envelope by binding to a GFP-nanobody (GFP-binding protein/GBP) appended to Heh2. Thus, without any perturbations to the nuclear envelope barrier, we can monitor the recruitment and activation of Chm7-GFP at the nuclear envelope, the latter of which we interpret as the local clustering of Chm7 as the most logical visual outcome of Chm7 polymerization at the level of fluorescence microscopy.

By shifting cells to medium containing galactose, we induced the expression of Chm7-GFP and monitored its distribution by timelapse microscopy. As shown in *Figure 4B*, Chm7-GFP was first observed in the cytosol but accumulated at the nuclear envelope within 20 min. Importantly, this nuclear envelope binding was due to its interaction with Heh2-GBP-mCherry as overexpression of Chm7-GFP in strains lacking Heh2-GBP-mCherry (*Figure 1C*) or lacking GBP (*Figure 4—figure supplement 1A,B*) did not lead to nuclear envelope accumulation or clustering. In contrast, Heh1-mCherry was incorporated into the Chm7-GFP-Heh2-GBP foci (*Figure 4—figure supplement 1C*). Interestingly, nearly simultaneously with the broader nuclear envelope-localization, Chm7-GFP and Heh2-GBP-mCherry accumulated in multiple foci throughout the nuclear envelope (see arrowheads at 40 min). These foci coalesced into one or two foci/cell over the length of the timecourse (90 min; *Video 1*). As a means to quantify this focal accumulation, we calculated a coefficient of variation (CV) of the mCherry fluorescence along the nuclear envelope in a mid-plane, which we plotted over time (*Figure 4C*). This approach faithfully represented the observed clustering, which reached a maximum value between 50 and 60 min (*Figure 4B,C*).

With the ability to temporally resolve recruitment and 'activation,' we next interrogated how Heh1 impacted these steps. Strikingly, the induction of Chm7-GFP expression in *heh1Δ* cells led to its accumulation at the nuclear envelope at a similar timepoint as in wildtype cells, however, we did not observe any focal accumulation with a CV remaining at ~1 over the 90 min timecourse (*Figure 4B,C*, and *Video 1*). Thus, Chm7 recruitment to the INM is not sufficient to lead to Chm7-GFP clustering. Instead, activation requires Heh1.

We next investigated whether other Chm7-interacting partners influenced Chm7-GFP clustering in the FETA assay beginning with Nur1. Interestingly, deletion of *NUR1* led to a statistically-significant drop in the CV of Heh2-GBP-mCherry at the end point of the FETA assay suggesting it could

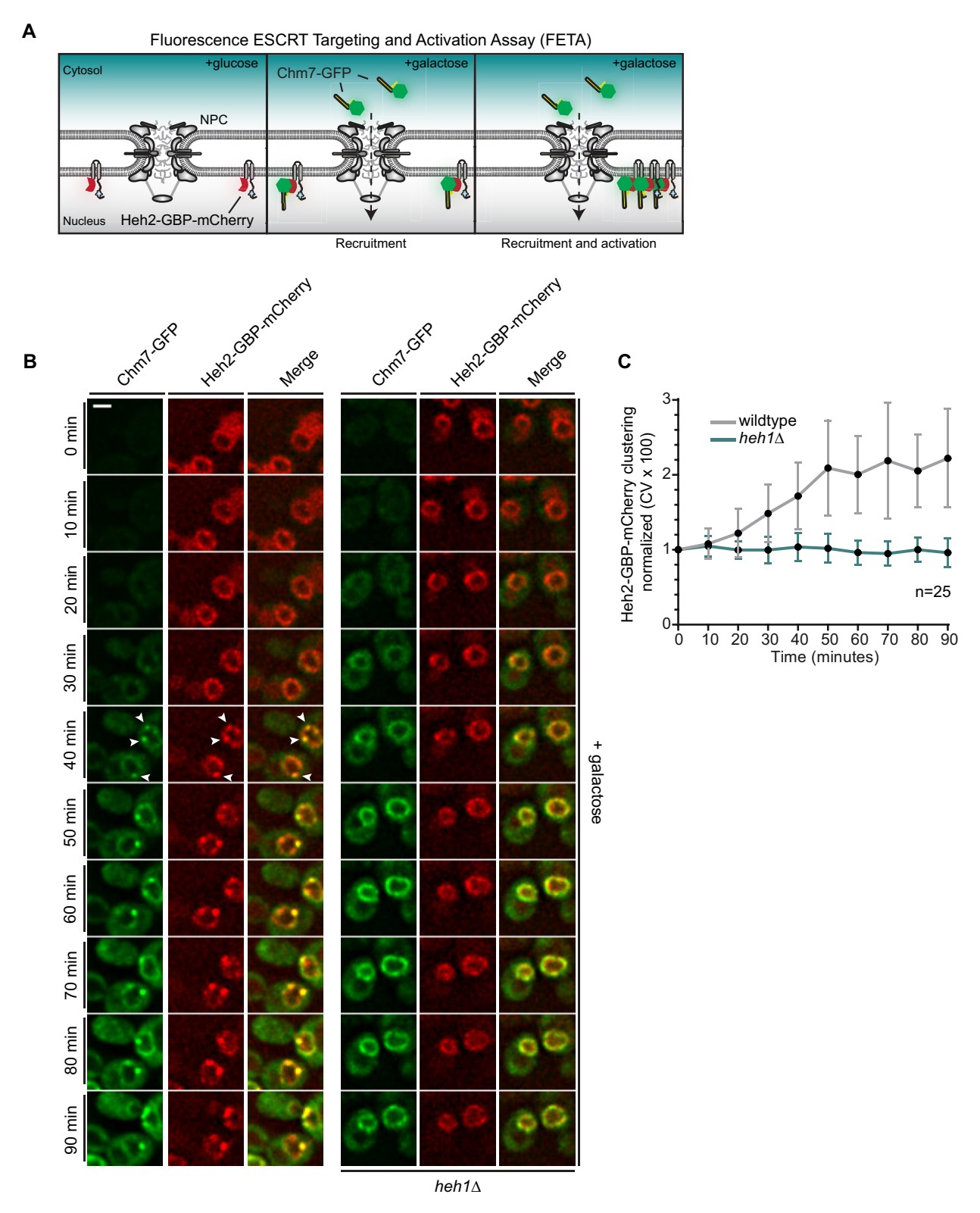

**Figure 4.** Heh1 is required to induce the focal accumulation of Chm7. (**A**) Schematic of 'Fluorescence ESCRT Targeting and Activation Assay' where Heh2 is expressed as a fusion to GFP-binding protein (GBP; red). Chm7-GFP is expressed by the addition of galactose to the growth medium. The focal accumulation of Heh2-GBP-mCherry is interpreted as Chm7 activation. (**B**) Deconvolved fluorescence micrographs of Chm7-GFP and Heh2-GBP-mCherry at the indicated timepoints after addition of galactose to the growth medium. Scale bar is 5 μm. (**C**) As a metric for the clustering of Heh2-

*Figure 4 continued on next page*

*Figure 4 continued*

GBP-mCherry, a coefficient of variation (CV) of the Heh2-GBP-mCherry fluorescence along the nuclear envelope in a mid-section was calculated over time. Mean and SD normalized to 0 timepoint are shown. n = 25.

DOI: https://doi.org/10.7554/eLife.45284.009

The following figure supplement is available for figure 4:

**Figure supplement 1.** Heh1 is required to induce the focal accumulation of Chm7.

DOI: https://doi.org/10.7554/eLife.45284.010

also contribute to Chm7 activation (*Figure 5A,B*). Consistent with this idea, we also observed considerably less chm7$_{OPEN}$-GFP accumulation at the nuclear envelope in *nur1Δ* cells (*Figure 5—figure supplement 1A,B*). However, we also noted that the total levels of Heh1 are reduced in *nur1Δ* cells (*Figure 5—figure supplement 1C*), suggesting that this effect may be indirect and ultimately mediated through Heh1. We further investigated the impact of deleting both *SNF7* and *VPS4* on the extent of Heh2-GBP-mCherry clustering in the FETA assay. In both cases, the CV of Heh2-GBP-mCherry was unaltered (*Figure 5B*). Yet, qualitatively, we observed that there were more discrete foci at the nuclear envelope in both of these genetic backgrounds, in addition to some foci (lacking Heh2-GBP-mCherry) in the cytoplasm (*Figure 5A*). Thus, these downstream components could impact other events that elude measurement and interpretation with this approach.

The lack of Heh2-GBP-mCherry clustering in *heh1Δ* cells also provided a genetic background to more fully vet the mechanism of Chm7 activation. First, introduction of *HEH1* on a plasmid rescued clustering of Chm7-GFP and Heh2-GBP-mCherry in the *heh1Δ* strains, confirming that lack of clustering was indeed due to the absence of Heh1 (*Figure 5C,D*). In contrast, expression of the *HEH1* paralogue, *HEH2*, failed to rescue clustering supporting that there are unique sequence elements in Heh1 that interface with Chm7 (*Figure 5—figure supplement 1D*). Consistent with this, expression of *heh1(1-735)*, which lacks the C-terminal WH domain, failed to restore Chm7 focal accumulation and Heh2-GBP-mCherry clustering suggesting that the WH domain was required for Chm7 activation (*Figure 5C,D*). Interestingly, however, the WH domain alone was insufficient to rescue clustering but instead it required a membrane anchor through a Heh1-transmembrane domain (*Figure 5C*, bottom panel). In the latter case, we also observed Chm7-GFP foci in the cytoplasm, likely in ER membranes, as the heh1(703-834) construct does not contain INM targeting sequences, consistent with data presented in *Figure 2A*. Thus, there is a clear coupling between the Heh1 WH domain and the membrane required for Chm7 activation.

## Chm7-Heh1 interactions drive nuclear envelope herniation and expansion

Our data support a model in which Chm7 and Heh1 are spatially segregated, but upon binding, Chm7 is locally activated at a membrane interface. To investigate how Chm7 activation could translate into a mechanism capable of sealing a nuclear envelope hole, we turned to a correlative light EM (CLEM) approach to investigate nuclear envelope morphology at sites of Chm7 activation. While our initial attempts focused on examining sites of Chm7-GFP localization in NPC assembly-defective strains that have nuclear envelope herniations like in *apq12Δ* and *nup116Δ* cells, the combination of the low abundance (and transience) of Chm7-GFP at these nuclear envelope foci coupled to the loss of GFP fluorescence through the freeze-substitution process precluded this as a viable approach. Thus, again, we turned to chm7$_{OPEN}$-GFP (and

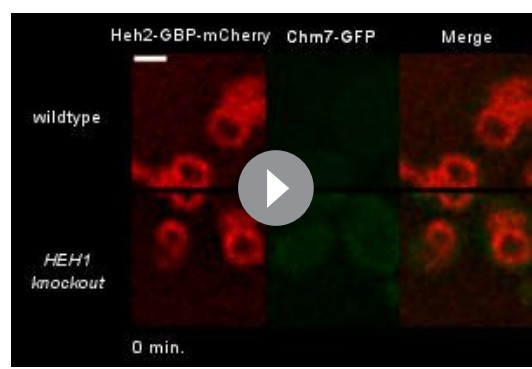

**Video 1.** Clustering of Heh2-GBP-mCherry in FETA assay requires Heh1. Related to *Figure 4*. A timelapse series of fluorescence images acquired at 10 min intervals of Chm7-GFP and Heh2-GBP-mCherry in wildtype and *heh1Δ* cells. Green, red and merge shown. Timestamp shows elapsed time after galactose induction of Chm7-GFP expression. Images were resized and pixels interpolated in FIJI. Scale bar is 2 μm.

DOI: https://doi.org/10.7554/eLife.45284.011

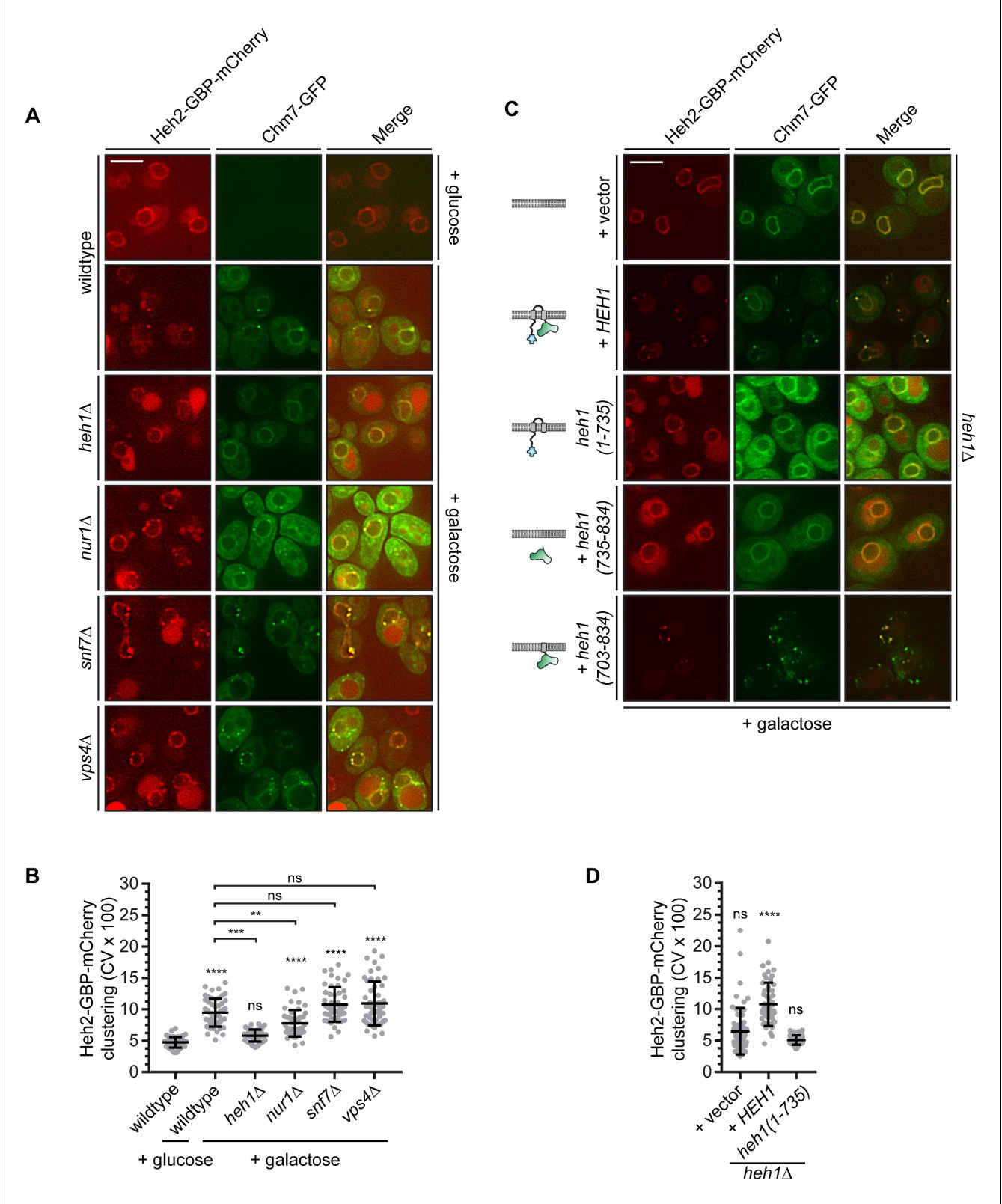

**Figure 5.** The Heh1 WH domain and a transmembrane anchor are necessary and sufficient for Chm7 activation, which is also modulated by Nur1, Snf7 and Vps4. (A) Deletion of *NUR1*, *SNF7* and *VPS4* impact Heh2-GBP-mCherry clustering in the FETA assay. Deconvolved fluorescence micrographs of Heh2-GBP-mCherry and Chm7-GFP in the indicated genetic backgrounds, either prior to (glucose) or after 90 min in galactose to drive Chm7-GFP production. Green and red channels with merge shown. Scale bar is 5 μm. (B) Plot of the CV of Heh2-GBP-mCherry fluorescence (as in *Figure 4C*) in the

*Figure 5 continued on next page*

*Figure 5 continued*

indicated strains before (glucose) or after 90 min of galactose addition to induce Chm7-GFP production. Data are from three independent replicates where 50 cells/genotype/replicate were counted. *P*-values are from two-way ANOVA with Dunnett's test where ns is p>0.05, **p≤0.01, ***p≤0.001, and ****p≤0.0001. (C) Deconvolved fluorescence micrographs of Heh2-GBP-mCherry and Chm7-GFP produced for 90 min in galactose. All images are from *heh1Δ* strains expressing the indicated genes encoding *HEH1* and several deletion constructs (schematized at left in a lipid bilayer). (D) Plot of the CV of Heh2-GBP-mCherry fluorescence (as in *Figure 4C*) in an *heh1Δ* strain expressing *HEH1* or *heh1(1-735)* after 90 min of galactose addition to induce Chm7-GFP production. Data are from three independent replicates of 50 cells per strain. *P*-values are from two-way ANOVA with Dunnett's test where ns is p>0.05, ****p≤0.0001.

DOI: https://doi.org/10.7554/eLife.45284.012

The following figure supplement is available for figure 5:

**Figure supplement 1.** The Heh1 WH domain and a transmembrane anchor are necessary and sufficient for Chm7 activation, which is also modulated by Nur1, Snf7 and Vps4.

DOI: https://doi.org/10.7554/eLife.45284.013

Chm7-GFP in *vps4Δpom152Δ* cells) as proxies to interrogate the membrane morphology at sites of surveillance (hyper) activation.

As shown in panel *i* of *Figure 6A and B*, we could effectively correlate fluorescence images and electron tomograms (*Kukulski et al., 2012*), which showed the chm7$_{OPEN}$-GFP foci apposed to the INM. Remarkably, this fluorescence demarked extensions of the INM that invade the nucleus and form a fenestrated network of INM cisterna – the lumen of the cisterna is continuous with the lumen of the nuclear envelope and is colored teal or purple to facilitate visualization. In *Figure 6A and B*, the panels *ii-iv* represent slices along the Z axis of the tomograms; 3D models were generated by isosurface rendering, which can be visualized as still frames (*Figure 6A,B,v–viii*) and in movies (*Videos 2* and *3*). These 3D views facilitate the observation of nuclear pores (denoted by stars) in addition to a perspective of the extent of the network of fenestrated cisternal membranes at the INM. Nearby, fenestrated membranes that we interpret to be ER also appeared with intriguing frequency (*Figure 6A,vii,viii*; *Figure 6B, iii*; 6 out of 14 tomograms).

The INM-associated cisternal network was often (11 out of 14 tomograms) found underneath balloon-like herniations of the nuclear envelope (i.e. both the INM and ONM) that extended several hundred nanometers into the cytosol (*Figure 6A,B*). The lumen of the herniations were open to the nucleoplasm and tapered into ~45 nm-diameter membrane 'necks': in the image shown in *Figure 6A,ii*, two 'necks' can be observed (black arrowheads). Often (in 10 out of 14 tomograms) vesicles appear nearby the sites of herniations some of which can be seen either fusing with, or fissioning from, the ONM (*Figure 6*, white arrowheads). Additional INM evaginations (extending into the nuclear envelope lumen, black arrowheads) are observed in tandem arrays that suggest they may be precursors of the herniations (*Figure 6A,iii*). Indeed, often several nuclear envelope herniations can be observed in a single tomogram, as indicated by white arrows in *Figure 6B*. Interestingly, these membrane deformations could also be formed in the absence of *SNF7* (*Figure 6—figure supplement 1A,B*, *Video 4*).

Similar membrane morphology was also observed when CLEM was applied to the focal accumulation of Chm7-GFP in *vps4Δpom152Δ* cells (*Figure 7A,i,*). Consistent with the idea that the INM expansion and nuclear envelope herniation need not occur simultaneously, in *Figure 7A* an example of a single herniation (with two necks; see black arrow heads) is shown. A nuclear-perspective (bottom-up, *Figure 7,v*) view allows a direct comparison between the herniation neck at the Chm7-GFP signal and nuclear pores that would be filled with NPCs (stars). Lastly, a more dramatic example of a *vps4Δpom152Δ* nuclear envelope where connections between the INM and a cisternal, lamellar membrane with multiple deformations is presented in *Figure 7B*. Interestingly, in this thick section, no nuclear envelope herniation is observed suggesting that there is no implicit link between the INM network and nuclear envelope herniation; these two morphologies might arise stochastically and are not necessarily directed in one, or the other, direction. Taken together, these data support a model in which the interaction of Chm7 and Heh1, and Chm7 activation, can lead to expansion of the INM and the formation of nuclear envelope herniations.

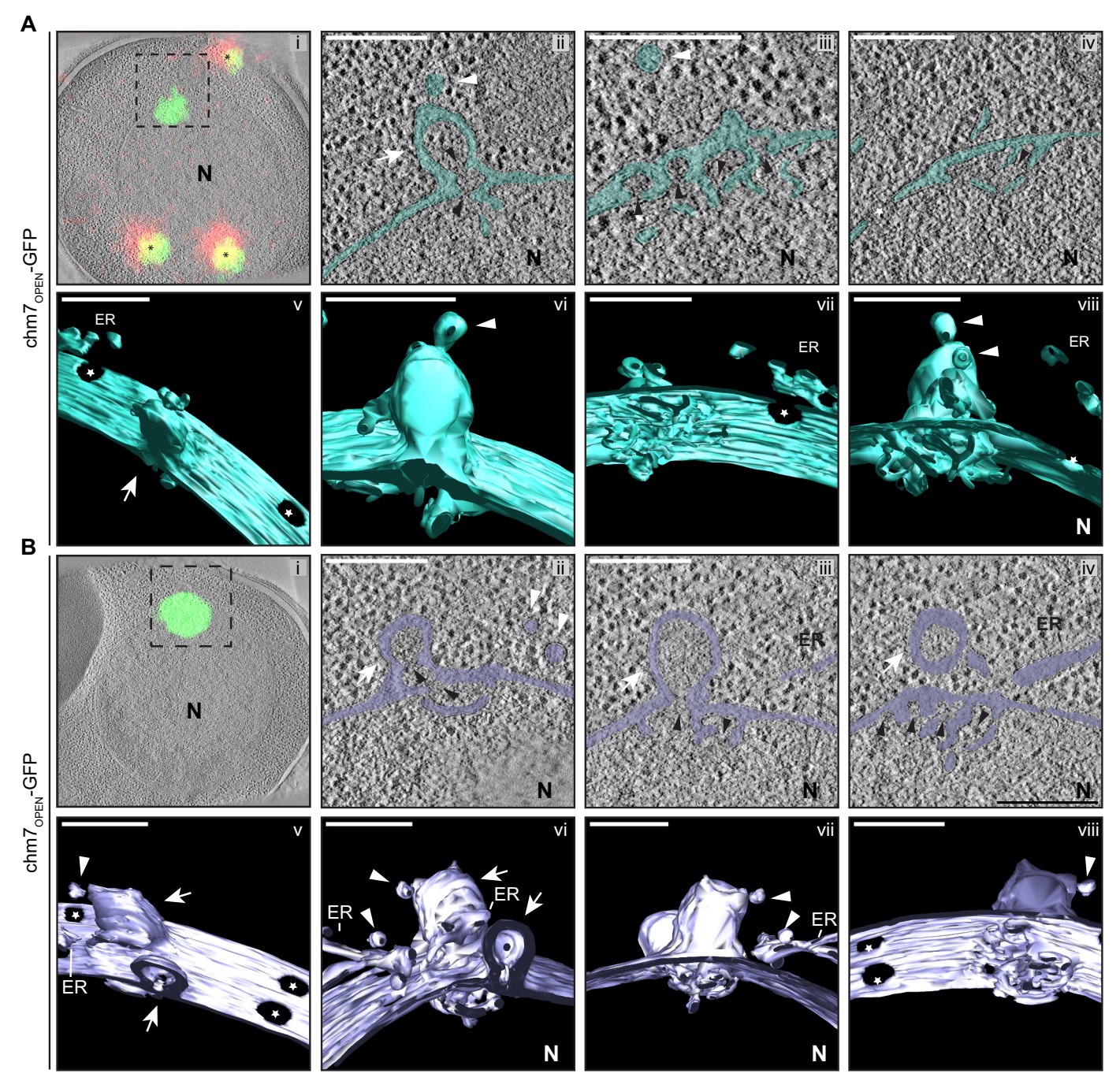

**Figure 6.** chm7$_{OPEN}$ associates with a network of intranuclear fenestrated cisterna often below nuclear envelope herniations. (**A, B**) Correlative light and electron microscopy of 300 nm thick sections was used to examine the morphology of the nuclear envelope at sites of chm7$_{OPEN}$-GFP accumulation. *i.* Overlay of fluorescent and electron micrographs with tetrafluorescent fiducials used for correlative alignment marked with (*); boxed region is magnified in *ii-iv*. *ii-iv*. Several views along the z-axis of the tomogram shown with the nuclear envelope/ER lumen filled with teal or light purple. *v-viii*. 3D models were generated and several perspective views are shown. White arrows are herniations, white arrowheads vesicles, black arrowheads are constrictions or necks of budding herniations, stars are nuclear pores. N is nucleus. Scale bars are 250 nm.
DOI: https://doi.org/10.7554/eLife.45284.014

The following figure supplement is available for figure 6:

**Figure supplement 1.** chm7$_{OPEN}$ associates with a network of intranuclear fenestrated cisterna often below nuclear envelope herniations.
DOI: https://doi.org/10.7554/eLife.45284.015

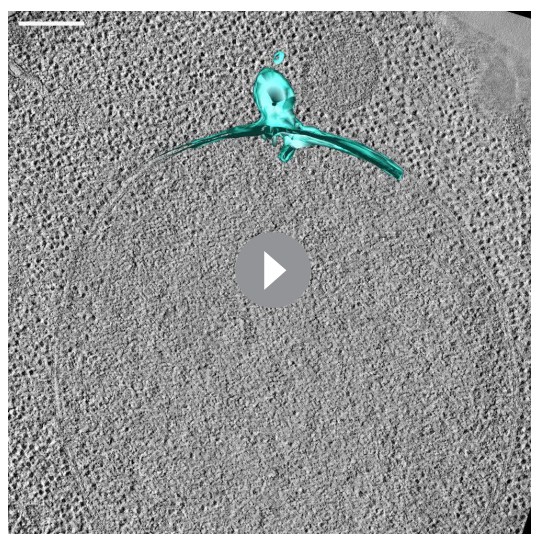

**Video 2.** Nuclear envelope morphology at sites of chm7_OPEN-GFP accumulation. Related to *Figure 6A*. Video showing full tomogram and 3D model from a nuclear envelope region of chm7_OPEN-GFP accumulation. Scale bar is 250 nm.
DOI: https://doi.org/10.7554/eLife.45284.016

## Morphologically distinct nuclear envelope herniations are associated with defects in NPC biogenesis

It was tempting to speculate that the nuclear envelope herniations that we observed under conditions of Chm7 activation were directly analogous to those observed in genetic backgrounds where NPC assembly is perturbed, like in *apq12Δ* and *nup116Δ* cells. To perform a direct comparison, we first confirmed that, as in *nup116Δ* cells (*Wente and Blobel, 1993*), NPC-like structures were found at the bases of the nuclear envelope herniations seen in cells lacking *APQ12* (*Scarcelli et al., 2007*) by staining thin sections with the MAb414 antibody that recognizes several FG-nups. As shown in *Figure 8A*, gold particles that label the MAb414 antibody were specifically found at the bases of these herniations confirming that they emanate from structures with nups. Furthermore, the diameter of the bases of these herniations averaged 78 nm, which while statistically similar to those found at mature NPCs (mean of 87 nm), were considerably larger than the ~45 nm diameter openings found at the necks of herniations caused by Chm7 (*Figure 8B*, *Video 5*). Lastly, the lumen of the herniations associated with both *apq12Δ* (*Figure 8A,C*) and *nup116Δ* cells (*Figure 8—figure supplement 1A*, *Video 6* and *Wente and Blobel, 1993*) are filled with electron density, whereas those associated with Chm7 appear to be empty (*Figure 6A,B*). Thus, we suggest that the herniations associated with overactive Chm7 and those associated with NPC assembly are morphologically distinct.

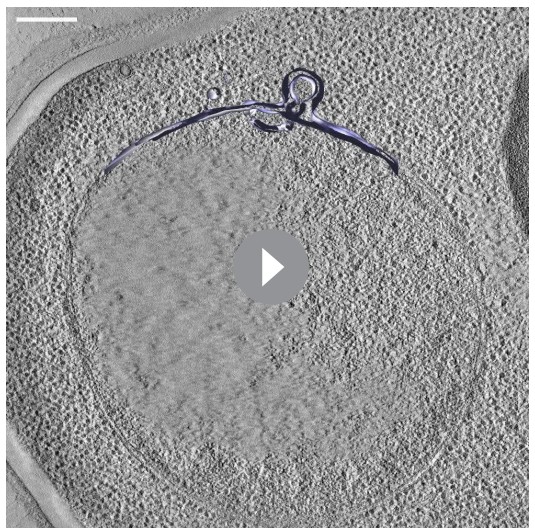

**Video 3.** Nuclear envelope morphology at sites of chm7_OPEN-GFP accumulation. Related to *Figure 6B*. Video showing full tomogram and 3D model from a nuclear envelope region of chm7_OPEN-GFP accumulation. Scale bar is 250 nm.
DOI: https://doi.org/10.7554/eLife.45284.017

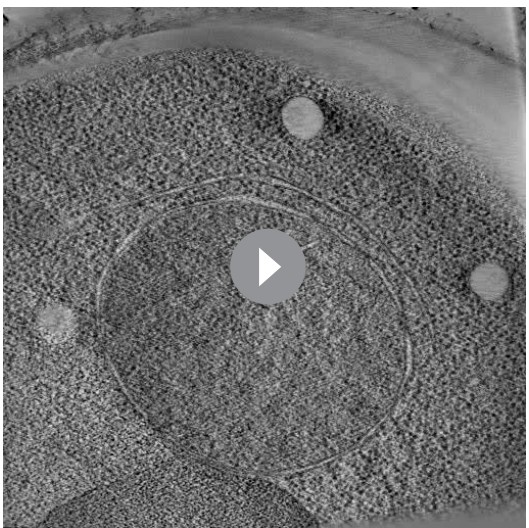

**Video 4.** Nuclear envelope morphology at sites of chm7_OPEN-GFP accumulation in the absence of *SNF7*. Related to *Figure 7A*. Video showing full tomogram and 3D model from a nuclear envelope region of chm7_OPEN-mCherry accumulation. Scale bar is 250 nm.
DOI: https://doi.org/10.7554/eLife.45284.018

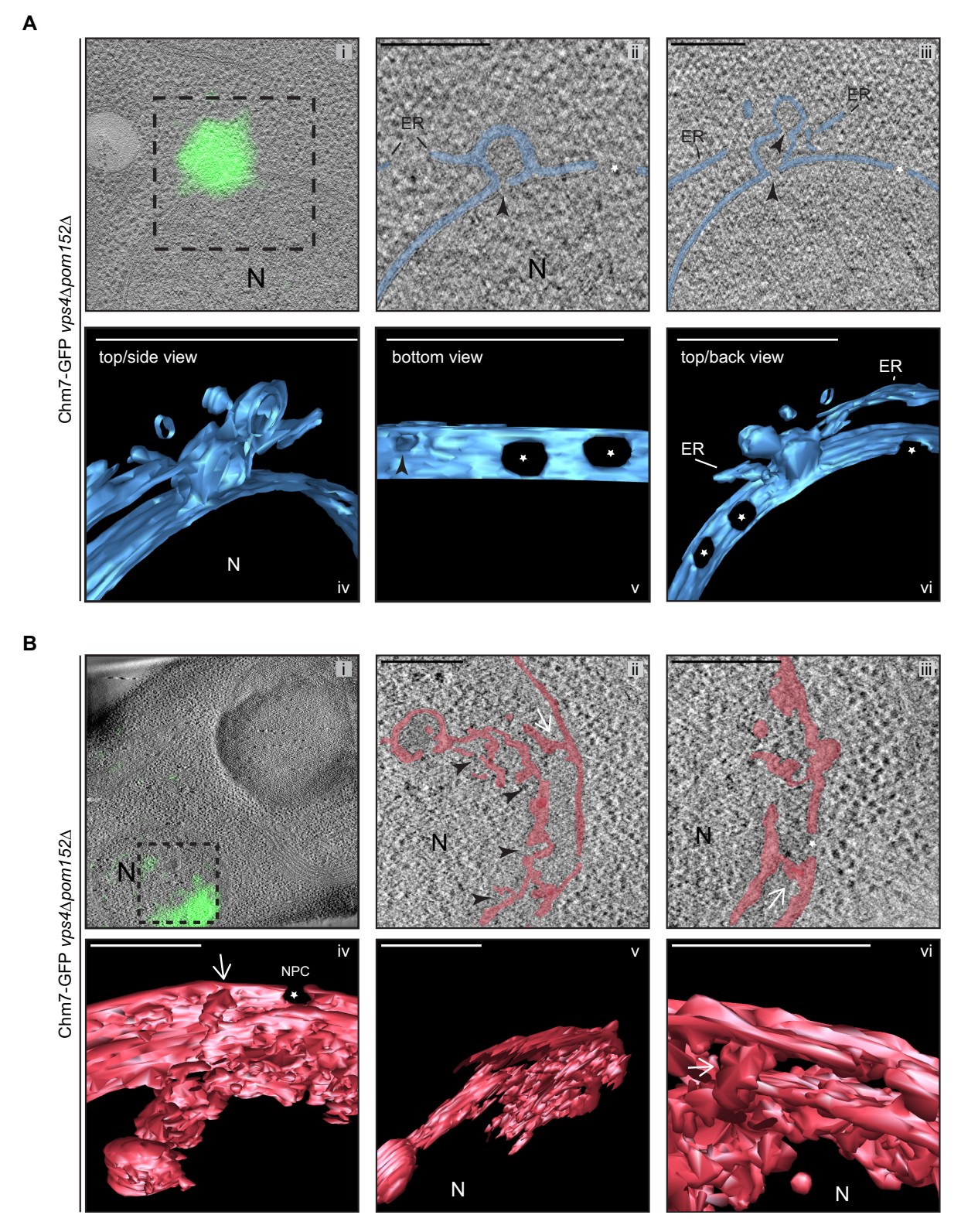

**Figure 7.** Chm7 associates with nuclear envelope herniations and intranuclear membranes. (A, B) Correlative light and electron microscopy of 300 nm thick sections was used to examine the morphology of the nuclear envelope at sites of Chm7-GFP accumulation in *vps4Δpom152Δ* cells. *i*. Overlay of fluorescent and electron micrographs; boxed region is magnified in *ii* and *iii*. *ii, iii*. Several views along the z-axis of the tomogram shown. *iv-vi*. 3D

*Figure 7 continued on next page*

*Figure 7 continued*

models were generated with indicated perspective views shown. Nuclear envelope/ER lumen colored blue or red. Black arrowheads are herniation necks or sites of membrane constriction, stars are nuclear pores. N is nucleus. Scale bars are 250 nm.

DOI: https://doi.org/10.7554/eLife.45284.019

That activated Chm7 might drive membrane expansion and nuclear envelope herniations with unique characteristics to those found in NPC assembly mutants does not exclude the possibility that Chm7 might nonetheless contribute to the formation of both of these herniation types. We therefore next investigated whether deletion of *CHM7* impacted the prevalence of herniations in *apq12Δ* cells. As shown in *Figure 8B and D*, deletion of *CHM7* had little impact on the number of herniations observed in thin sections of *apq12Δ* cells, which were only modestly reduced (23% versus 31% of nuclei; *Figure 8B*). Most strikingly, however, we observed that 35% of the nuclei in the thin sections of *apq12Δchm7Δ* cells (*Figure 8B,E* and *Figure 8—figure supplement 2*) had large (>500 nm) discontinuities in their nuclear membranes, which suggests that these nuclei were unstable and could rupture (labeled as nuclear envelope rupture/NER). Similar nuclear envelope discontinuities were observed in *apq12Δsnf7Δ* strains (*Figure 8—figure supplement 3*). Indeed, in some cases we could observe nucleoplasm escaping into the cytosol (*Figure 8E*, left panel). This result provides an explanation for the striking loss of NLS-GFP reporter accumulation in the nucleus that was observed in only ~35% of *apq12Δchm7Δ* cells (*Webster et al., 2016*). Thus, while these NPC-assembly-associated nuclear envelope herniations might not require *CHM7* or *SNF7* for their biogenesis, ESCRTs are nonetheless required to maintain the integrity of the nuclear membranes in the context of these herniations.

## Discussion

While there has been considerable focus over the last few decades on mechanisms that control the targeting of proteins and lipids to distinct intracellular compartments, it is equally important to understand the protective mechanisms that maintain this compartmentalization in the face of challenges to membrane integrity and/or the specific biochemical identity of organelles. Here, we further explore the mechanism of ESCRT surveillance of the nuclear envelope. We interpret our data in a model where the nuclear envelope is surveilled by two principle components, the ESCRT Chm7 and the integral INM protein, Heh1. This surveillance system appears to be set up to respond directly to perturbations in the nuclear envelope barrier in a way that we suggest is agnostic as to whether the perturbation is a result of defectively formed NPCs or a mechanical (or other) disruption of the nuclear membranes.

The rationale behind this assertion is that the nuclear envelope surveillance system is itself directly established by a functioning nuclear transport system, which physically segregates Chm7 and Heh1 on either side of the nuclear envelope. For example, prior work has shown that Heh1 requires the function of the NTRs Kap-α and Kap-β1 in addition to the Ran-GTPase in order to be actively targeted to the INM through NPCs (*King et al., 2006*). Here, we establish that Chm7, while small enough to passively leak through the NPC diffusion barrier, is actively exported by the major export NTR, Xpo1 (*Figure 1*). Therefore, any perturbations that impact active nuclear transport or the diffusion barrier across the nuclear envelope would lead to increased Chm7 diffusion into the nucleus and/or a deficit in its nuclear export increasing the likelihood that it meets Heh1. While similar perturbations could also lead to Heh1 mistargeting and/or its diffusion into the ONM, we suspect that this would be kinetically slower than Chm7 diffusion into the nucleus as Heh1 is bound to chromatin at the INM (*Grund et al., 2008*; *Mekhail et al., 2008*; *Gonzalez et al., 2012*; *Yam et al., 2013*; *Barton et al., 2015*; *Schreiner et al., 2015*). Indeed, it is probable that Heh1 functions in two major roles with respect to nuclear integrity: first, it provides mechanical stability to the nucleus by binding chromatin, and second, provides a binding site for Chm7 through its C-terminal WH domain. These two inter-related roles could also help to explain why both the N- and C-terminal domains of Heh1 are required to maintain viability of *apq12Δ* cells (*Figure 2C*), and why *HEH1* is generally more essential in budding and fission yeasts compared to *CHM7*. This likely holds true in mammalian models as well as LEM2 is essential whereas CHMP7 is dispensable for viability (in cell culture) (*Hart et al., 2015*).

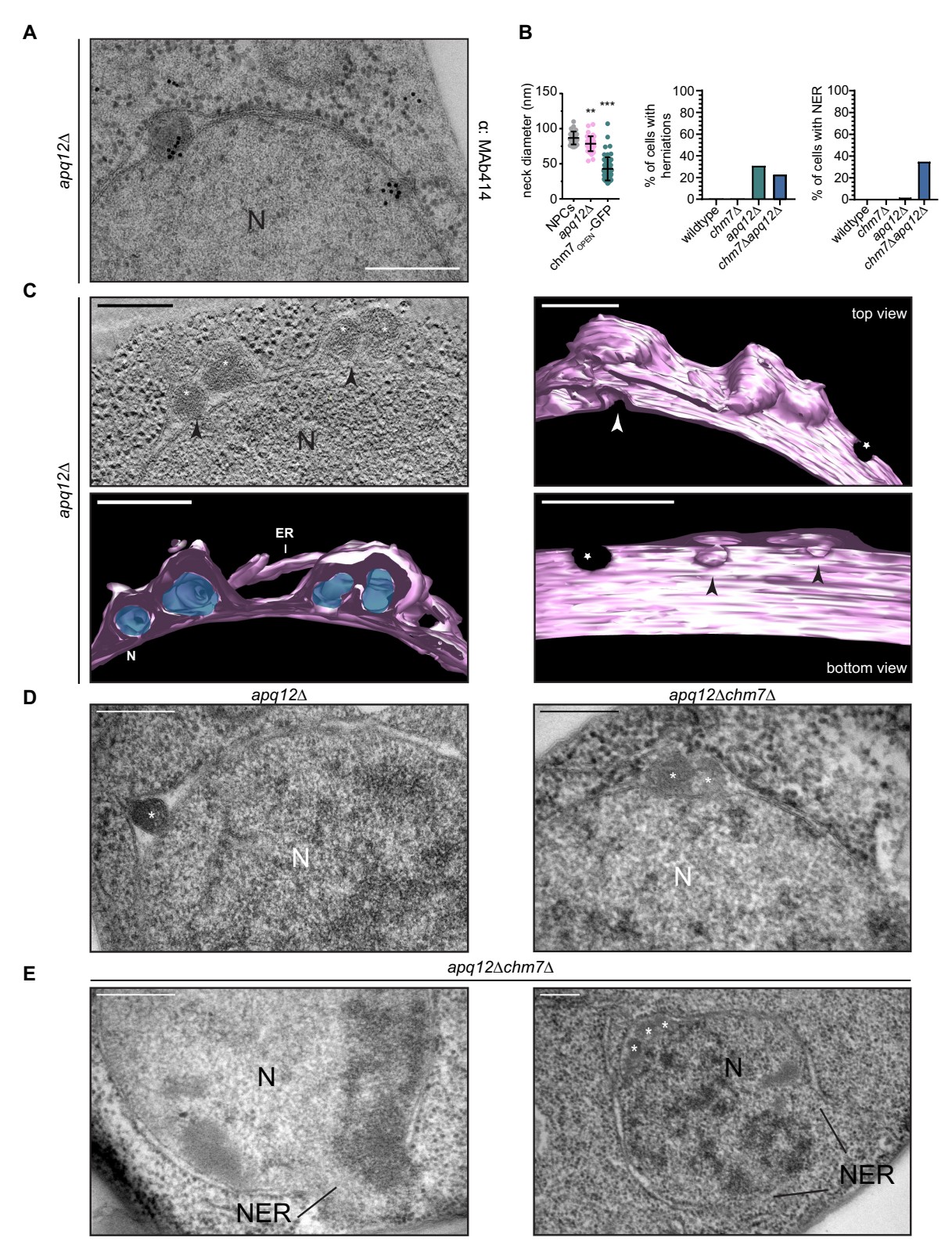

**Figure 8.** *CHM7* is required to maintain the integrity of the nuclear membranes in the context of nucleoporin-associated herniations. (**A**) Nucleoporins are found at the bases of nuclear envelope herniations in *apq12Δ* cells. Immunogold labelling of thin sections of *apq12Δ* cells with 5 nm gold-conjugated secondary antibodies that detect MAb414 labeled nups at bases of herniations. (**B**) Left: Plot of the diameter of herniation necks in the indicated genetic backgrounds and fenestrations within the intranuclear membrane network associated with chm7_OPEN. Middle: Plot of the percentage

*Figure 8 continued on next page*

*Figure 8 continued*

of nuclei of the indicated strains where nuclear envelope herniations are observed. Right: Plot of the percentage of nuclei in the indicated strains where nuclear envelope ruptures (NER) are observed. At least 100 cells from each genotype were quantified. *P*-values are from Student's t-test, where **p≤0.01, ***p≤0.001. (C) Electron tomograph of 300 nm thick section of *apq12Δ* cells grown at 37°C for 2 hr. Note the electron density within the herniations. Perspective views of 3D model shown (membranes/nuclear envelope-ER lumen colored pink with electron density within herniation blue). Arrowheads point to herniation necks and stars are nuclear pores. Scale bars are 250 nm. (D) Nuclear envelope herniations associated with nups persist in the absence of *CHM7*. Representative electron micrographs of the *apq12Δ* and *chm7Δapq12Δ* strains grown for 2 hr at 37°C. Asterisks denote herniation lumen. N is nucleus. (E) Nuclear envelope ruptures (NER) are observed in *chm7Δapq12Δ* cells. Electron micrographs of *chm7Δapq12Δ* depicting nuclear envelope ruptures (NER). Nucleus is indicated with 'N'. Scale bars are 250 nm.

DOI: https://doi.org/10.7554/eLife.45284.020

The following figure supplements are available for figure 8:

**Figure supplement 1.** *CHM7* is required to maintain the integrity of the nuclear membranes in the context of nucleoporin-associated herniations.
DOI: https://doi.org/10.7554/eLife.45284.021
**Figure supplement 2.** *CHM7* is required to maintain the integrity of the nuclear membranes in the context of nucleoporin-associated herniations.
DOI: https://doi.org/10.7554/eLife.45284.022
**Figure supplement 3.** *CHM7* is required to maintain the integrity of the nuclear membranes in the context of nucleoporin-associated herniations.
DOI: https://doi.org/10.7554/eLife.45284.023

Once a perturbation occurs to the nuclear envelope barrier through defects in NPC assembly, loss of function of NPCs, or mechanical disruption of the nuclear membranes, Chm7 and Heh1 are able to come together. While we have so far been unsuccessful in reconstituting a direct biochemical interaction between Chm7 and Heh1 (although others have between LEM2 and CHMP7 [*Gu et al., 2017*]), we have nonetheless provided evidence that this interaction likely leads to Chm7 activation, thus tightly coupling recruitment and activation (*Figure 4*). Our data support a model in which the C-terminal WH domain of Heh1 in addition to a membrane anchor are necessary and sufficient for this activation event. As our prior work (*Webster et al., 2016*) and that from *Olmos et al. (2016)*, support that it is the N-terminal ESCRT-II-like domain of Chm7 (which, interestingly, is also predicted to be made up of tandem WH domains [*Horii et al., 2006*; *Bauer et al., 2015*; *Figure 1A*]) that is necessary for recruitment to the nuclear envelope, it seems likely that the binding between the Heh1-WH domain and the N-terminus of Chm7 could trigger activation by removing some form of autoinhibition, which is a common theme among ESCRT-III proteins (*Zamborlini et al., 2006*; *Shim et al., 2007*; *Lata et al., 2008*; *Bajorek et al., 2009*; *Henne et al., 2012*; *Tang et al., 2015*; *Tang et al., 2016*) that ensures polymerization in the correct compartment. Clearly, the precise molecular mechanism of Chm7 activation by Heh1 will require structural insight, which is no doubt on the horizon.

Even with structural information, there are many additional factors that need to incorporated into a nuclear envelope surveillance mechanism. For example, future work must directly address how Heh2 fits into this pathway. While Heh2 has a similar domain architecture as Heh1 including a WH domain (*King et al., 2006*), it does not appear to impact Chm7 activation within the FETA assay (*Figure 5—figure supplement 1D*), nor was it detected in any of the affinity purifications (*Figure 3*). This was surprising, as we had previously established that the N-terminus of Heh2 (that lacks the WH domain), can directly bind to both Chm7 and Snf7, at least in their 'open' forms (*Webster et al., 2016*). Moreover, we reported physical interactions between Chm7 and Snf7 with Heh2 in vivo (*Webster et al., 2014*; *Webster et al., 2016*). While there are many possible explanations for these results, we favor models that consider Heh2 in a regulatory role that might modulate Chm7 and/or Snf7 function at the nuclear envelope, although this remains speculative and awaits direct experimentation.

Additional downstream (of Chm7 and Heh1) components also need to be figured into any nuclear envelope surveillance mechanism including Snf7, but also Did2, Vps24 and Vps2 (*Figure 3B,C*). That these ESCRT-IIIs impact the recruitment of Vps4 to the nuclear envelope (*Figure 3E*) raises the possibility that the sequence of their recruitment might be similar to that found at ILVs during MVB formation, which begins with Snf7 and ends with Did2, Vps24, and Vps2 (*Babst et al., 2002*; *Teis et al., 2008*), the latter of which play important roles in Vps4 recruitment and ATPase activation required for membrane scission (*Azmi et al., 2008*; *Shestakova et al., 2010*; *Schöneberg et al., 2018*). Interestingly, however, while we observe Vps4 recruitment to sites of chm7$_{OPEN}$ at the INM (*Figure 3E*),

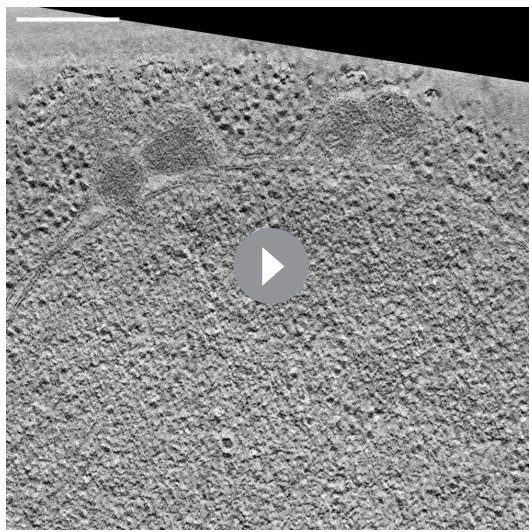

**Video 5.** Morphology of nuclear envelope herniations in *apq12Δ* cells. Related to *Figure 8C*. Video showing a tomogram and 3D model of the nuclear envelope in *apq12Δ* cells. Scale bar is 250 nm.
DOI: https://doi.org/10.7554/eLife.45284.024

we do not have any explicit evidence for membrane scission per se. Indeed, the INM evaginations and nuclear envelope herniations appear to be stabilized by ~45 nm diameter necks (*Figures 6*, *7* and *8B*). That these necks have similar diameters to those observed at the plasma membrane or endosomes when HIV budding (*von Schwedler et al., 2003*; *Morita et al., 2011*; *Cashikar et al., 2014*; *Jackson et al., 2017*) or ILV formation (*Adell et al., 2014*; *Buono et al., 2017*; *Frankel et al., 2017*; *Wenzel et al., 2018*), respectively, are stalled, is suggestive that they are stabilized by a similar ESCRT-III-like polymer. We suggest that the most likely explanation is that Chm7 itself might play a unique role in a nuclear membrane-scission step, perhaps through its potential MIM1 domain that might directly recruit Vps4 but is absent from the chm7$_{OPEN}$ construct. Such a conclusion is supported by the presence of similar INM evaginations and nuclear envelope herniations observed in the presence of full length Chm7 but lacking Vps4 (*Figure 7*).

Any scission mechanism that requires a direct interaction between the MIM1 domain of Chm7 and Vps4 will also need to consider the overlap of this sequence with NES2 (*Figure 1*). This clearly predicts a competition between Xpo1 and Vps4 for Chm7, which provides a potential entryway for an additional level of regulation for any polymer formation or scission reaction by this NTR (that would also be modulated by Ran-GTP). Indeed, NTRs might regulate this pathway at the level of Heh1 as well. Heh1 is synthesized and inserted into ER membranes so is exposed to the cytosol and would thus be capable of binding to Chm7 before it is targeted to the INM. Such an assumption is supported by the observation that the exposure of the WH domain of Heh1 is sufficient to recruit Chm7 to ER membranes (*Figure 2*). Does NTR-binding to the NLSs in the N-terminus of Heh1 inhibit Chm7 recruitment in this compartment? Answering this question will also be relevant in mammalian systems where LEM2 and CHMP7 are found in the same compartment during mitotic nuclear envelope breakdown suggesting the need for mechanisms to prevent their binding in this phase of the cell cycle. Interestingly, recent work is suggesting that proteins like Lgd/CC2D1B help control the spatiotemporal timing of CHMP4B and CHMP7 activity in mammalian cells (*Ventimiglia et al., 2018*), supporting the existence of regulatory mechanisms of this pathway.

Lgd/CC2D18 acts during nuclear envelope reformation when ESCRTs play critical roles in sealing the nuclear envelope at the end of mitosis, often at sites where spindle microtubules perforate the nascent nuclear envelope (*Olmos et al., 2015*; *Olmos et al., 2016*; *Vietri et al., 2015*). It will be interesting to understand the similarities and differences

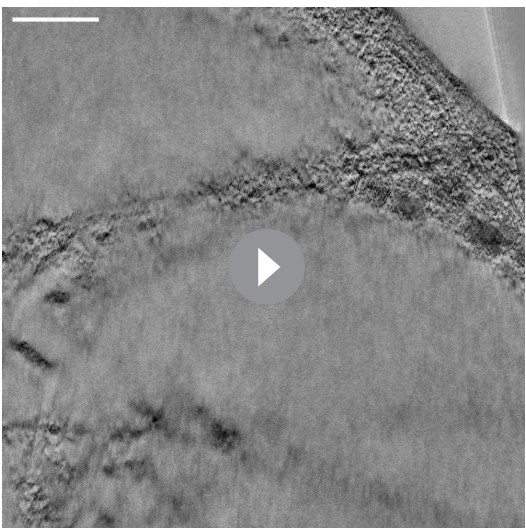

**Video 6.** Morphology of nuclear envelope herniations in *nup116Δ* cells. Related to *Figure 8—figure supplement 1A*. Video showing a tomogram and 3D model of the nuclear envelope in *nup116Δ* cells. Scale bar is 250 nm.
DOI: https://doi.org/10.7554/eLife.45284.025

between how ESCRTs function at the end of mitosis (in an open mitosis) versus events that are triggered upon acute disruption of nuclear envelope integrity or NPC function during interphase. For example, two of the more interesting observations of the EM tomographic analyses are the generation of intranuclear membrane invaginations and the proximity of ER sheets and vesicles at the ONM sites of nuclear envelope herniations (*Figure 6*, *Figure 6—figure supplement 1*, *Figure 7*). Both of these observations suggest that nuclear envelope surveillance might be coupled to the delivery of new membrane at sites of rupture. It seems plausible that sealing holes in the nuclear envelope (particularly those larger than a nuclear pore) would require the local delivery of membranes. Similarly, NPC quality control mechanisms that suggest the sealing of defective NPCs would likely require some form of expansion of the pore membrane, as has been proposed (*Wente and Blobel, 1993*). Whether such membrane is derived from new synthesis or from the mobilization of existing stores remains to be explored. Of note, recent work supports that even the INM may be metabolically active and have the capacity to generate new lipid locally (*Romanauska and Köhler, 2018*). Further, close examination of sites of chm7$_{OPEN}$ accumulation reveals that the morphology of the INM extensions are both sheet-like and tubular in nature and resemble the 'normal' continuities between the ONM and the broader ER network (*West et al., 2011*); together these data suggest that ER-shaping proteins and lipid synthesis pathways might also play an important role in contributing to nuclear envelope sealing - topics for the future.

Both ER remodeling proteins and lipid synthesis pathways are required for de novo NPC biogenesis (*Schneiter et al., 1996*; *Dawson et al., 2009*; *Hodge et al., 2010*). As NPC biogenesis proceeds through an INM evagination step, we also must consider whether the INM evaginations observed in the electron tomograms reflect a function for Chm7 in this process. However, while this remains a compelling hypothesis, our data do not yet support this idea. For example, the herniations observed in the context of inhibiting NPC assembly are morphologically distinct from those found in the chm7$_{OPEN}$ or *vps4Δpom152Δ* scenarios with wider necks that contain nups (*Figure 8A,B*). Further, they arise in the absence of *CHM7* and are thus more likely to be formed due to a defect in NPC biogenesis (perhaps because of an inhibition of INM-ONM fusion), or, through the triggering of a NPC assembly surveillance mechanism (*Webster et al., 2016*; *Thaller and Lusk, 2018*). Should Chm7 be a critical component of the latter mechanism, it would be predicted that the nuclear envelope herniations associated with nups would be unsealed without *CHM7*. Indeed, that we observe often-dramatic openings in the nuclear envelope in *apq12Δchm7Δ* cells (*Figure 8D* and *Figure 8—figure supplement 2*), suggest that the herniations themselves might be prone to rupture with Chm7 required for their repair. It follows then that like in mammalian cells where much larger nuclear envelope herniations are precursors to nuclear rupture (*De Vos et al., 2011*; *Vargas et al., 2012*; *Hatch et al., 2013*; *Denais et al., 2016*; *Hatch and Hetzer, 2016*; *Raab et al., 2016*), these smaller NPC-assembly associated herniations might also impact nuclear envelope integrity through mechanisms that remain to be fully defined. In either case, it reinforces the concept that the assembly of NPCs can be perilous, and it will be important to consider this possibility when interpreting the underlying pathology of human diseases that are associated with defects in NPC function or assembly, for example, DYT1 early-onset dystonia (*Laudermilch et al., 2016*; *Pappas et al., 2018*) or Steroid Resistant Nephrotic Syndrome (*Miyake et al., 2015*; *Braun et al., 2016*; *Braun et al., 2018*).

Lastly, while we acknowledge then that the INM evaginations that we observe in cases of Chm7 hyperactivation might not necessarily be a physiological event in a nuclear envelope sealing process, it remains tempting to speculate that they might be in the context of proposed mechanisms of nuclear egress be it Mega-RNPs (*Speese et al., 2012*; *Jokhi et al., 2013*), viruses (*Lee et al., 2012*; *Lee et al., 2016*; *Arii et al., 2018*), or nucleophagy (*Roberts et al., 2003*; *Dou et al., 2015*; *Mochida et al., 2015*; *Mostofa et al., 2018*) that so-far remain obscure but would nonetheless require a membrane scission step. Interestingly, recent work suggests that herpes virus nuclear egress requires ESCRTs (*Arii et al., 2018*), including a role in controlling INM extensions into the nucleus; such intranuclear membrane might also be relevant in cell types that have so-called 'nucleoplasmic reticulum' (*Malhas et al., 2011*). The biogenesis and function of nucleoplasmic reticulum remain enigmatic but our observations of intranuclear fenestrated membrane emanating from the INM might suggest a yet-to-be discovered role for Chm7 and Heh1 in forming such structures as well.

# Materials and methods

## Yeast strains and growth conditions

All strains used in this study are from a W303 parent; their derivation and genotypes are listed in *Supplementary file 1*. Fluorescent protein tagging and gene deletions were generated using a PCR-based integration approach using the pFA6a plasmid series (*Supplementary file 2*) as templates (*Longtine et al., 1998*; *Van Driessche et al., 2005*). Standard yeast protocols for transformation, mating, sporulation, chromosomal DNA isolation and tetrad dissection were followed (*Amberg et al., 2005*).

Cells were grown to mid-log phase in YPA (1% Bacto yeast extract (BD), 2% Bacto peptone (BD), 0.025% adenine hemi-sulfate (Sigma)) or complete synthetic medium (CSM) supplemented with 2% raffinose (R; BD), 2% D-galactose (G; Alfa Aesar) or 2% D-glucose (D; Sigma) as indicated.

To compare relative growth rates of *heh1Δapq12Δ* strains expressing *HEH1* alleles (DTCPL1498, DTCPL1517, DTCPL1581, DTCPL1519, DTCPL1520) roughly equivalent cell numbers from overnight cultures grown in in YPAR were spotted in 10-fold serial dilutions onto YPG to induce expression of Heh1 or indicated truncations and imaged after 36 hr at 30°C.

## Leptomycin B and BAPTA-AM treatments

To test the impact of inhibiting nuclear export on the steady state distribution Chm7-GFP, Chm7-MGM4, chm7$_{OPEN}$-GFP, GFP, NES1$_{CHM7}$-GFP, NES2$_{CHM7}$-GFP, or NES1-2$_{CHM7}$-GFP, we used KWY175 (a gift from B. Montpetit and Karsten Weis) in which the genomic deletion of *XPO1* is covered with pRS413 expressing the *xpo1-T539C* allele that confers sensitivity to Leptomycin B (LMB). These strains were grown in YPAR and galactose (final concentration of 1%) was added to the growth medium to induce the expression of the GFP fusion proteins for 2 hr before the addition of 2% D-glucose to repress protein production. Cultures were then treated with 50 ng/mL LMB dissolved in 7:3 MeOH:H$_2$O solution (Sigma) for 45 min alongside a control of the equivalent volume of MeOH before imaging.

To test if Ca$^{2+}$ plays a role in the physiological recruitment of Chm7 to the nuclear envelope, *apq12Δ* cells expressing Chm7-GFP (DTCPL567) were cultured overnight at 30°C, diluted to an OD$_{600}$ of 0.2 and grown for an additional 2 hr at RT. Cells were treated with either 25 μM (*Li et al., 2011*) of the cell permeable calcium chelater BAPTA-AM (Tocris Bioscience) dissolved in DMSO or DMSO alone for 30 min followed by a 45 min incubation at either RT or 37°C before imaging.

## Plasmids

All plasmids are listed in *Supplementary file 2*.

To generate pRS406-ADH1-GFP, the GFP coding sequence was amplified by PCR and inserted into pRS406-ADH1 (p406ADH1 was a gift from Nicolas Buchler and Fred Cross - Addgene plasmid # 15974; http://n2t.net/addgene:15974; RRID:Addgene_15974) using *Eco*RI and *Hind*III restriction sites.

To generate pCHM7-MGM4, the *CHM7* ORF was amplified by PCR with *Cla*I restriction sites and subcloned in pPP004 linearized with *Cla*I (gift of L Veenhoff; *Popken et al., 2015*), placing *CHM7* in between the first MBP gene and the GFP.

To generate pRS406-ADH1-NES1-GFP a Geneblock (IDT) was synthesized with the coding sequence for amino acids 370–390 of Chm7 with homology arms containing 40 base pairs flanking sequences outside of the multiple cloning site in pRS406-ADH1-GFP. The geneblock was assembled into pRS406-ADH1-GFP using the Gibson Assembly reaction (New England BioLabs).

To generate pRS406-ADH1-NES2-GFP, complimentary 4 nmol Ultramers (IDT) were designed to code for amino acids 409–424 of Chm7 with overhangs that would be generated with *Xba*I or *Bam*HI. Ultramers were annealed by heating to 95°C for 5 min in Taq polymerase PCR buffer (Invitrogen) and allowing them to slowly cool to RT. Annealed primers were then ligated using T4 ligase (Invitrogen) into pRS406-ADH1-GFP linearized with *Xba*I/*Bam*HI (New England BioLabs).

To generate pRS406-ADH1-NES1-2$_{CHM7}$-GFP, the sequence encoding amino acids 370–429 of Chm7 was amplified with oligonucleotide primers containing the *Xba*I or *Bam*HI restriction sites. The PCR product was digested with *Xba*I/*Bam*HI and gel purified (Qiagen) before ligation with T4 ligase (Invitrogen) into pRS406-ADH1-GFP linearized with *Xba*I/*Bam*HI.

Gibson Assembly (New England BioLabs) was used to generate pFA6a-3xHA-GFP-his3MX6 for functional tagging of Vps4 (*Adell et al., 2017*). The 3xHA epitope was PCR-amplified from a pFA6a-3xHA-his3MX6 (*Longtine et al., 1998*) plasmid using Q5 DNA polymerase (New England BioLabs) and assembled into pFA6a-GFP-his3MX6 (*Longtine et al., 1998*) digested with SalI and PacI (New England BioLabs).

## Western blotting

For whole cell protein extracts, approximately 2 $OD_{600}$ of cells in mid log phase were collected by centrifugation, washed in 1 mM EDTA, pelleted again and resuspended/lysed in 2 M NaOH for 10 min on ice. Proteins were precipitated by the addition of 50% Trichloroacetic acid for 20 min on ice and then collected by centrifugation. The precipitated proteins were washed in ice-cold acetone, air dried and then resuspended in SDS-PAGE sample buffer. Samples were then denatured at 95°C for 5 min. Denatured proteins were separated on precast SDS-PAGE, 4–20% gradient gels (BioRad) and transferred to 0.2 µm nitrocellulose membranes (BioRad). Relative protein loading was visualized using Ponceau S Solution (Sigma). Membranes were subsequently washed in TBST and blocked for 1 hr in 5% skim milk in TBST at RT. Membranes were then incubated with HRP-conjugated anti-HA (Roche 3F10), or anti-actin (mAbcam 8224) diluted in TBST. Primary antibodies were detected directly with ECL (ThermoFisher) or with anti-rabbit HRP-conjugated secondary antibodies, followed by ECL and visualized using a VersaDoc Imaging System (Bio-Rad).

## Fluorescence microscopy

With the exception of the correlative light electron microscopy experiments described below, all fluorescence micrographs were acquired using a DeltaVision microsope (Applied Precision/GE Healthcare) fitted with a 100x, 1.4 NA objective (Olympus). Images were taken using a CoolSnapHQ$^2$ CCD camera (Photometrics), with the exception of those in *Figure 2* and *Figure 4B*, which were acquired using a Evolve EMCCD camera (Photometrics).

For timecourse assessment of Heh2-GBP-mCherry clustering in the FETA assay in *Figure 4B*, cells were imaged in microfluidic plates (Y04C/CellASIC) in the ONIX microfluidic platform (CellASIC). Cells were loaded into the microfluidic chamber in CSM with 2% raffinose. CSM with 2% galactose was perfused into the microfluidic chamber at 0.25 psi for the course of the experiment. Z-stacks (0.4 µm sections) were acquired for 90 min at 10 min intervals.

## Image processing and analysis

All presented fluorescent micrographs were deconvolved using an iterative algorithm in softWoRx (6.5.1; Applied Precision GE Healthcare). Unprocessed images after background subtraction were used for quantification of fluorescence intensities.

Assessment of Heh2-GBP-mCherry or Heh2-mCherry clustering was quantified by calculating the coefficient of variation (SD/mean x 100) of fluorescence of individual nuclear envelopes (*Fernandez-Martinez et al., 2012*). A four pixel wide, freehand line was traced over the entire nuclear envelope in a mid-plane section using FIJI/ImageJ (*Schindelin et al., 2012*) and the mean fluorescence contained in the traced area was measured. Cells with vacuolar autofluorescence that obscured fluorescence at the nuclear envelope were excluded from quantification. Further, in some *heh1Δ* cells Heh2-GBP-mCherry was found in plaque-like clusters before Chm7-GFP induction. Thus, as this phenotype was not triggered by expression of Chm7-GFP, they were excluded from the clustering analysis.

To correlate the fluorescence intensity of co-localized Vps4-GFP and chm7$_{OPEN}$-mCherry, the integrated density of Vps4-GFP and chm7$_{OPEN}$-mCherry was measured and plotted on a correlation curve. The linear correlation coefficient (Pearson coefficient, r) was calculated in Prism (GraphPad 8.0).

Similarly, quantification of the integrated density and average fluorescence intensity of Vps4-GFP and chm7$_{OPEN}$-mCherry were measured by selecting a region of interest (ROI) around the chm7$_{OPEN}$-mCherry signal and measuring average fluorescence intensity in both mCherry and GFP channels.

To measure relative nuclear exclusion of GFP, NES1$_{CHM7}$-GFP, NES2$_{CHM7}$-GFP, and NES1-2$_{CHM7}$-GFP constructs at steady state, a 3.75 µm line was traced across each nucleus (encompassing

cytoplasm both times the line crosses the nuclear envelope border) as determined from the dsRed-HDEL localization. GFP fluorescence was measured using the Plot Profile function in FIJI/ImageJ (*Schindelin et al., 2012*). Traces were normalized to the maximum value measured within each trace before averaging.

## Statistical analyses

Graphs and statistical analyses were generated using Prism (GraphPad 8.0). *P*-values in all graphs were generated with tests as indicated in figure legends and are represented as follows: ns, p>0.05; *p≤0.05; **p≤0.01 ***p≤0.001, ****p≤0.0001. All error bars represent the standard deviation from the mean. Scatter plots of spectral counts from MS/MS analysis for *Figure 3A,B and C* were generated using Excel (Microsoft).

## Nuclear export signal prediction

Xpo1/Crm1 NES sequences were predicted using LocNES (*Xu et al., 2015*) with default threshold settings.

## Recombinant protein-binding experiments

GST, GST-Chm7 and GST-heh1(735-834) (the Heh1 WH domain) proteins were recombinantly produced and purified as previously described (*Webster et al., 2016*) in lysis buffer (50 mM Tris pH 7.4, 500 mM NaCl, 2 mM $MgCl_2$, 2 mM $CaCl_2$,10% glycerol, 0.5% NP-40, 1 mM DTT, complete protease inhibitors (Roche)). The soluble fraction was incubated with glutathione sepharose (GT) beads for 1 hr at 4°C for binding. The GT beads were collected by centrifugation and washed thrice with lysis buffer. GST and GST-Chm7 proteins were eluted from GT beads by 10 mM reduced glutathione and dialyzed in lysis buffer. The Heh1 WH was cleaved from GT beads by incubating with HRV3C protease (Thermo Scientific) at 4°C overnight. For binding experiment, Heh1 WH was incubated with GST and GST-Chm7 in a dialysis cassette (3.5K Slide-A-Lyzer, Thermo Scientific) and the binding reaction was dialyzed overnight in a binding buffer (50 mM Tris pH 7.4, 150 mM NaCl, 2 mM $MgCl_2$, 2 mM $CaCl_2$,10% glycerol, 0.5% NP-40, 1 mM DTT). The reaction was collected, incubated with GT beads for 1 hr at 4°C, washed thrice with binding buffer and eluted in 2X Laemmli sample-buffer. Proteins were resolved on a SDS-PAGE gel and visualized by SimplyBlue Safe-Stain (Invitrogen).

## Immunoaffinity purification

*S. cerevisiae* cells expressing Chm7-GFP (DTCPL81), Chm7-GFP *vps4Δpom152Δ* (DTCPL133), chm7$_{OPEN}$-GFP (DTCPL413) were grown to log phase in YPD at 30°C and collected by centrifugation. Cells were washed once with ice-cold water, collected by centrifugation and resuspended in a small volume of freezing buffer (20 mM HEPES, pH 7.4, 1.2% polyvinylpyrrolidone and protease inhibitor [Sigma; *Oeffinger et al., 2007*]) and flash frozen in liquid nitrogen. The frozen yeast pellets were pulverized in a Retsch MM400 mixer mill for 6 times at 30 Hz for 3 min. For immunoaffinity purification, 200 mg of frozen, ground yeast powder was solubilized in 4 volumes of homogenization buffer (400 mM trisodium citrate, pH 8, 0.5% n-Dodecyl β-D-maltoside) and protease inhibitor cocktail (Roche). The soluble fraction was incubated with 10 μl magnetic beads (Dynabeads, M-270 Epoxy, Invitrogen) slurry coated with GFP-nanobody for 1 hr at 4°C (*Cristea et al., 2005*; *LaCava et al., 2016*). The beads were collected on a magnetic rack and washed three times with 500 μl homogenization buffer. Bound proteins were eluted by incubating the beads in 20 μl 1X NuPAGE LDS (lithium dodecyl sulfate) sample buffer (Invitrogen) at 70°C for 10 min. Eluates were separated on a magnetic rack and further incubated with 50 mM DTT at 70°C for 10 min. The eluates were run on a 4–12% NuPAGE gel (Novex) until the dye front just entered the gel. The gels were stained with Imperial protein stain (Thermo Scientific) and a protein band (consisting of all eluted proteins) were excised for MS analysis.

## Mass spectrometry and analysis

MS/MS was performed at the Yale Keck Proteomics facility. Excised bands described above were transferred to clean 1.5 mL Eppendorf tubes and digested with trypsin. Subsequently, chromatographic separation of peptides was done using a Waters nanoACQUITY ultra high pressure liquid chromatograph (UPLC), and peptides were detected on a Waters/Micromass AB QSTAR Elite.

Analysis of MS/MS peptide results was completed using Scaffold 4.8.7 (Proteome Software Inc). Peptides were identified by SEQUEST and Mascot using X!Tandem (*Craig and Beavis, 2003*; *Searle et al., 2008*) and validated using PeptideProphet (*Keller et al., 2002*; *Nesvizhskii et al., 2003*) within Scaffold software (Proteome Software Inc). Proteins were identified by comparison with SwissProt database where peptide identifications required ≥2 peptides from each replicate and ≥95.0% probability of correct identification to be included in analysis. Quantitative analysis to determine significance of enrichment between samples was done with total spectral counts from two replicates using Fischer's exact test with a significance threshold $p < 0.05$ (*Figure 3A,C*), or on presence/absence from one replicate (*Figure 3B*).

## Correlative fluorescence and electron tomography

Correlated fluorescence and electron microscopy were conducted as previously described (*Kukulski et al., 2012*; *Curwin et al., 2016*). In brief, yeast cells were high pressure frozen (HPM010, AbraFluid), freeze substituted (EM-AFS2, Leica) with 0.1% uranyl acetate in acetone and infiltrated with Lowicryl. 300 nm sections were cut with a microtome (EM UC7, Leica) and picked up on carbon coated 200 mesh copper grids. 50 nm TretraSpeck fluorescent microspheres (fluorescence and electron dense fiducials, Life technologies, Carlsbad, CA) were added to the grid for correlation. Grids used for *Figure 6A,B*, *Figure 6—figure supplement 1A,B*, *Figure 8C*, *Figure 8—figure supplement 1A*, and *Figure 8—figure supplement 3A*, were poststained with lead citrate to increase contrast. In all cases, 15 nm protein A-coupled gold beads were adsorbed on both sides of each grid and used as fiducial markers for overlaying high and low magnification tomograms. 60° to −60° tilt series were acquired on a Technai F30 (Thermofisher, FEI) at 300 kV with Serial-EM (*Mastronarde, 2005*) at 20000x and either 3900x or 4700x to facilitate ease of correlation with TetraSpeck fiducials.

To perform CLEM, fluorescence images were acquired of the EM grids on images a Nikon TI-E (*Figure 6A,B*) with sCMOS PCO edge 4.2 CL camera and solid state illumination, or an Olympus IX81 with MT20 (Olympus) lamp and CCD (Orca-ER; Hamamatsu Photonics) (*Figure 7A,B*, *Figure 6—figure supplement 1A,B*). To distinguish protein fluorescence signal from fluorescent fiducials, for each field of view/grid four channels were acquired (GFP, mCherry/RFP, Cy5, and brightfield). Acquired images were further processed in FIJI using the Extended Depth of Field Plugin (*Forster et al., 2004*). Correlation of fluorescence and reconstructed electron tomograms was performed using the ec-CLEM Plugin (*Paul-Gilloteaux et al., 2017*) in ICY (*de Chaumont et al., 2012*). Alignment was determined by clicking on corresponding pairs of TetraSpeck fiducials in the two imaging modalities.

Tomograms were reconstructed using the IMOD package (Windows Version 6.2) and Etomo (Version 4.9.8, *Kremer et al., 1996*). Patch tracking function was used to perform a fiducial-less image alignment for reconstruction. 3DMOD software was used for manual segmentation of the tomograms. Further editing and annotation were done in Adobe Illustrator (Adobe). Video sequences were compiled in 3DMOD and exported with further editing in ImageJ/FIJI (*Schindelin et al., 2012*). Video frames were compressed as JPGs to reduce file size.

## 2D electron microscopy

To examine the ultrastructure of *apq12Δ* (CPL1326) and *apq12Δchm7Δ* (CPL1327) strains, unfixed cells were high-pressure frozen using a Leica HMP100 at 2,000 psi and freeze-substituted using a Leica Freeze AFS unit using 1% osmium tetroxide and 1% glutaraldehyde. Samples were infiltrated with durcupan resin (Electron Microscopy Science) and cut in 100 nm thick sections using a Leica UltraCut UC7. Sections were collected on formvar/carbon coated nickel grids and stained with 2% uranyl acetate and lead citrate. Grids were imaged in a FEI Tecnai Biotwin TEM at 80 kV with a Morada CCD camera and iTEM (Olympus) Software.

## Immunogold labeling

For immunogold labeling of nucleoporins, 70 nm Lowicryl sections generated as described above for correlative light and electron tomography were cut using Leica UltraCut UC7 onto 200 mesh copper grids (Quantifoil Micro Tools GmbH). Immunolabeling was carried out with the MAb414 antibody diluted 1:100 in 1% BSA, followed by washes in PBS, and probing with a secondary 10 nm gold-

conjugated antibody. After further washes, the grids were fixed in 1% glutaraldehyde in PBS. Lastly, grids were post-stained with 1% uranyl acetate, washed in water and viewed with a Biotwin CM120 Philips equipped with a 1K CCD Camera (Keen View, SIS).

## Acknowledgments

We thank members of the Lusk, Beck and King laboratories for critical input in addition to M King for comments on the manuscript. We also acknowledge invaluable assistance from Z Hakhverdyan and M Rout for affinity purifications and the Yale Keck Proteomics facility, M Graham and X Liu for help with EM and generous support by EMBL's electron microscopy core facility and Y Schwab. This work was supported by the NIH, GM105672 to CPL and DJT was also funded by 5T32GM007223 and a short term EMBO fellowship 6885. MA was funded by an EMBO a long term fellowship (ALTF-1389–2016).

## Additional information

### Funding

| Funder | Grant reference number | Author |
|---|---|---|
| European Molecular Biology Organization | Fellowship 6885 | David J Thaller |
| National Institutes of Health | 5T32GM007223 | David J Thaller |
| European Molecular Biology Organization | Fellowship ALTF-1389-2016 | Matteo Allegretti |
| National Institutes of Health | GM105672 | C Patrick Lusk |

The funders had no role in study design, data collection and interpretation, or the decision to submit the work for publication.

### Author contributions

David J Thaller, Conceptualization, Formal analysis, Validation, Investigation, Visualization, Methodology, Writing—original draft, Writing—review and editing; Matteo Allegretti, Formal analysis, Validation, Investigation, Visualization, Writing—review and editing; Sapan Borah, Investigation, Visualization, Writing—review and editing; Paolo Ronchi, Resources, Investigation, Writing—review and editing; Martin Beck, Resources, Supervision, Project administration, Writing—review and editing; C Patrick Lusk, Conceptualization, Resources, Formal analysis, Supervision, Funding acquisition, Validation, Investigation, Visualization, Methodology, Writing—original draft, Project administration, Writing—review and editing

### Author ORCIDs

David J Thaller http://orcid.org/0000-0003-3577-5562
Martin Beck http://orcid.org/0000-0002-7397-1321
C Patrick Lusk http://orcid.org/0000-0003-4703-0533

### Decision letter and Author response

Decision letter https://doi.org/10.7554/eLife.45284.030
Author response https://doi.org/10.7554/eLife.45284.031

## Additional files

### Supplementary files

• Supplementary file 1. List of all yeast strains used in this study, their source and/or derivation.
DOI: https://doi.org/10.7554/eLife.45284.026

• Supplementary file 2. List of all plasmids used in this study, their source and/or derivation.
DOI: https://doi.org/10.7554/eLife.45284.027

• Transparent reporting form
DOI: https://doi.org/10.7554/eLife.45284.028

**Data availability**

All data generated and analyzed during this study are included in the manuscript and supporting files. Source data files have been provided for Figure 3.

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
