## [Decision Letter]

Thank you for submitting your work article "An ESCRT-LEM protein surveillance system is poised to directly monitor the nuclear envelope and nuclear transport system" for consideration by *eLife*. Your article has been reviewed by three peer reviewers, one of whom is a member of our Board of Reviewing Editors, and the evaluation has been overseen by Vivek Malhotra as the Senior Editor.

The reviewers have discussed the reviews with one another and the Reviewing Editor has drafted this decision to help you prepare a revised submission.

Thaller et al. address the question of how the nuclear envelope (NE) recognizes and repairs damage that has occurred as a result of mechanical tears of the NE or as a result of mis-assembly of the nuclear pore complex (NPC). The authors present evidence that nuclear transport maintains the protein Heh1 on the nuclear side of the NE, and Chm7 – a component of the ESCRT-III membrane remodeling and scission complex – in the cytoplasm. The authors' previous work (Webster et al., 2014 and 2016) established the link between Chm7 and Heh1. In the present manuscript, Heh1 is shown to be required for Chm7 accumulation at nuclear membrane foci, which is taken as an indication of Chm7 activation at the NE – the term 'activation' implies Chm7 polymerization within the larger framework of ESCRT-III, but the nature of this activation beyond accumulation of Chm7 at foci is not developed. A novel light microscopy assay developed by the authors shows that disruption of the NE or of NPC function allows Heh1 and Chm7 to meet locally at the site of disruption and initiate local membrane remodeling. Collectively, the data suggest that recruitment of Chm7 to the sites of Heh1 exposure and subsequent ESCRT-III-dependent membrane sealing represents a surveillance mechanism to repair NE damage and restore nucleocytoplasmic integrity.

Overall this is an outstanding contribution that was broadly supported by the reviewers. The data are of excellent quality, the authors develop and implement a useful new assay, and the conclusions give insight into the different membrane structures that form in response to NE perturbation. Most importantly, the findings support a mechanistic model that will guide further inquiry into this poorly understood cellular mechanism. We have the following suggestions to improve the manuscript:

1) The title focuses on LEM, but the action is via WH domains. Consider dropping "LEM domain" from the title and/or working "WH domain" into the title in some manner.

2) By fusing NES2 and the whole NES1-NES2 region to GFP, the authors conclude that the NES2 alone is not sufficient for efficient export, whereas the full NES1-NES2 region is export-proficient. As described in the Discussion, a potential MIM domain overlaps the NES2 in Chm7. Because MIMs are sites in ESCRT-III proteins that bind associated factors (most notably, the Vps4 ATPase that disassembles ESCRT-III polymers), the MIM-NES2 overlap raises the possibility that binding of other proteins to the Chm7 MIM could influence the binding of Xpo1? An experiment to assess if the NES1 is sufficient to exclude Chm7 from the nucleus would be important to shed light on this question.

3) The MIM motif in Chm7 was described in Bauer et al., 2015. The authors should cite this paper specifically when they describe this motif (paragraph 2 of the Results section).

4) In Figure 1—figure supplement 1A, please cite a reference for the consensus sequence of the MIM motif.

5) In Figure 3, interaction partners of the mutant Chm7(open) protein are described from pulldowns. Heh1 is the top hit (along with ESCRT-III components), but Heh2 is not found as an interaction partner. However, the authors' previous work in Webster et al., 2016, established that the N-terminal domain of Heh2 (a homolog of Heh1) can directly interact with the Chm7 C-terminal domain in vitro. Why is Heh2 not detected as a binding partner in vivo? Are Heh2 levels too low to detect; is there no in-vivo binding possible; or does Heh2 never meet Chm7? Additionally, in Figure 5 the authors show that the WH domain of Heh1 is required for the activation of Chm7. Because Heh2 also contains a WH domain, can we assume that (theoretically) Heh2 is also able to activate Chm7? The authors can do a better job discussing this.

6) The authors should take the opportunity here to expand upon the likely relationship between the NE herniations they observe and those implicated by the Budnik lab in no-NPC mediated RNA export? There is some significant confusion in the field on this point.

7) Figure 3D, bottom row: please indicate more clearly the channel are we looking at, red or green.

8) Figure 3E: Vps60 and Ist1 mutants were also tested according to the figure but are not mentioned anywhere in the text. Why were they chosen and what do the authors conclude from the microscopy?

9) Figure 5: the caption does not cover the contents of panel A and B of this figure at all. Please, rephrase the caption.

10) Subsection “Chm7 binds to an INM platform”, last paragraph – about Vps20: for clarity, add a sentence about why, in vps20-deleted cells, Vps4-GFP is no longer in cytosolic puncta and has increased levels at the Chm7(open) locus. Is it because Vps20 is the endosomal binding partner of Vps4, and by deleting it, more Vps4 is available for binding at the NE?

11) Subsection “Fluorescence ESCRT Targeting and Activation (FETA) Assay”, last paragraph – Figure 5C and D: When Heh1(703-834) is expressed, Chm7-GFP is cytosolic. Is it located at the ER?

12)"there are obvious parallels between morphologies at INM, plasma membrane and endosomes." To make the Discussion an easier read, please explain these obvious parallels in a bit more detail. What was observed in these cited papers?

13) Discussion, fourth paragraph – please correct: accumulation of Chm7 in vps4d pom152d cells, not vps4d only.

14) Discussion, fifth paragraph: in the first line it is stated that the size of the necks at evaginations and herniations is suggestive of the presence of a spiraling polymer as observed in vitro and in vivo. To clarify this statement, it would be good to make explicit what McCullough et al. observed and with what proteins.

15) Typo, "where the mechanism (of) Chm7 recruitment and activation can be decoupled."

16) How is ER distinguished from outer nuclear membrane herniations in tomograms? The authors imply ER membrane as a source involved in repair.

17) If Chm7 is not required for the NE membrane remodeling (as seen in apq12- chm7-double deleted cells) how could the open-mutant active form of Chm7 on its own induce the membrane dynamics seen by tomography? In essence, Chm7 is not needed for this event to occur, but Chm7 induces it. Somehow, that logic needs to be fleshed out.

18) The accumulation of Chm7 at foci in Figure 1C would be more obvious with arrowheads noting a couple of these instances.

19) Why is the bottom row of images in Figure 1C displayed differently than the 2 upper rows?

20) The conclusion regarding the NLS of Lem2 in Kralt et al. is not as described in the manuscript. Kralt et al. found that the NLS of Lem2 is NOT like those of Heh1 or Heh2. Only that of Pom121 bears similarity.

---

## [Author Response]

[…] Overall this is an outstanding contribution that was broadly supported by the reviewers. The data are of excellent quality, the authors develop and implement a useful new assay, and the conclusions give insight into the different membrane structures that form in response to NE perturbation. Most importantly, the findings support a mechanistic model that will guide further inquiry into this poorly understood cellular mechanism. We have the following suggestions to improve the manuscript:1) The title focuses on LEM, but the action is via WH domains. Consider dropping "LEM domain" from the title and/or working "WH domain" into the title in some manner.

We seriously considered altering the title to better reflect the role of the WH domain. However, we have opted to retain “LEM protein” as this is a better recognized nomenclature for this family of conserved integral inner nuclear membrane proteins and thus we felt it would ultimately prove to be more attractive to the general reader.

2) By fusing NES2 and the whole NES1-NES2 region to GFP, the authors conclude that the NES2 alone is not sufficient for efficient export, whereas the full NES1-NES2 region is export-proficient. As described in the Discussion, a potential MIM domain overlaps the NES2 in Chm7. Because MIMs are sites in ESCRT-III proteins that bind associated factors (most notably, the Vps4 ATPase that disassembles ESCRT-III polymers), the MIM-NES2 overlap raises the possibility that binding of other proteins to the Chm7 MIM could influence the binding of Xpo1? An experiment to assess if the NES1 is sufficient to exclude Chm7 from the nucleus would be important to shed light on this question.

Yes, we completely agree that the overlap between NES2 and potential MIM domain raises the possibility of direct competition between Xpo1 and Vps4 or other factors; this is addressed in the Discussion. We were unable to assess this possibility experimentally, however, because we simply do not understand what factors (including Vps4) actually bind to the MIM domain. Moreover, while deletion of NES1 is an important experiment, this experiment in isolation would not do justice to what is undoubtedly a complex interplay between NES1, NES2 and the MIM1 domain, which requires more in-depth study using both in vivo and in vitro tools that we feel is beyond the scope of the current work. We did recognize, however, that the figure exploring the sufficiency of the NESs was incomplete as it did not formally establish that NES1, on its own, could act as an NES. These data are now presented in a revised Figure 1F.

3) The MIM motif in Chm7 was described in Bauer et al., 2015. The authors should cite this paper specifically when they describe this motif (paragraph 2 of the Results section).

Thank you for ensuring this work is properly referenced.

4) In Figure 1—figure supplement 1A, please cite a reference for the consensus sequence of the MIM motif.

The reference has now been added in the figure legend.

5) In Figure 3, interaction partners of the mutant Chm7(open) protein are described from pulldowns. Heh1 is the top hit (along with ESCRT-III components), but Heh2 is not found as an interaction partner. However, the authors' previous work in Webster et al., 2016, established that the N-terminal domain of Heh2 (a homolog of Heh1) can directly interact with the Chm7 C-terminal domain in vitro. Why is Heh2 not detected as a binding partner in vivo? Are Heh2 levels too low to detect; is there no in-vivo binding possible; or does Heh2 never meet Chm7? Additionally, in Figure 5 the authors show that the WH domain of Heh1 is required for the activation of Chm7. Because Heh2 also contains a WH domain, can we assume that (theoretically) Heh2 is also able to activate Chm7? The authors can do a better job discussing this.

These are all excellent questions and ones that we have been asking for some time now. Ultimately, we think that the answers will have to come from in vitro reconstitution experiments that can directly tackle the potential competition and/or functional overlap between the Heh1 and Heh2 WH domains (and also their N-termini). In the meantime, we now present data in a **revised** Figure 5—figure supplement 1D, that Heh2 is unable to activate Chm7 in the FETA assay. Thus, there is no functional overlap between the WH domains of Heh1 and Heh2 at least with respect to Chm7 activation. We dedicate a portion of the Discussion to addressing how Heh2 might fit into this pathway, which we more effectively place in the context of our prior data.

6) The authors should take the opportunity here to expand upon the likely relationship between the NE herniations they observe and those implicated by the Budnik lab in no-NPC mediated RNA export? There is some significant confusion in the field on this point.

We agree that there is some confusion in the field regarding these observations by the Budnik group. While our data certainly support the concept that ESCRTs might be capable of budding RNPs into the lumen, we have not explicitly explored this possibility experimentally and therefore cannot be definitive for or against any such non-NPC transport model. Therefore, while we certainly consider this as a possibility in the Discussion, I worry about adding further confusion to the field if we overreach.

7) Figure 3D, bottom row: please indicate more clearly the channel are we looking at, red or green.

We have now added sufficient detail to the figure legend to address this point.

8) Figure 3E: Vps60 and Ist1 mutants were also tested according to the figure but are not mentioned anywhere in the text. Why were they chosen and what do the authors conclude from the microscopy?

Thank you for pointing out that these were not discussed, we have now included a rationale for their inclusion in the Results.

9) Figure 5: the caption does not cover the contents of panel A and B of this figure at all. Please, rephrase the caption.

This caption and figure legend are now more explicit as to the contents of the figure.

10) Subsection “Chm7 binds to an INM platform”, last paragraph – about Vps20: for clarity, add a sentence about why, in vps20-deleted cells, Vps4-GFP is no longer in cytosolic puncta and has increased levels at the Chm7(open) locus. Is it because Vps20 is the endosomal binding partner of Vps4, and by deleting it, more Vps4 is available for binding at the NE?

Essentially yes, deletion of *VPS20* should prevent the recruitment of downstream ESCRTs and Vps4 to endosomes potentially freeing up a larger pool of Vps4 to bind at the nuclear envelope. We have now revised this section of the Results to address this point.

11) Subsection “Fluorescence ESCRT Targeting and Activation (FETA) Assay”, last paragraph – Figure 5C and D: When Heh1(703-834) is expressed, Chm7-GFP is cytosolic. Is it located at the ER?

As this construct has a transmembrane domain but lacks targeting information to the INM, it is our assumption that it is predominantly found in the ER. It would, however, be small enough to diffuse into the INM and thus still be capable (as shown) of activating Chm7 and thus clustering Heh2-GBP-mCherry.

12)"there are obvious parallels between morphologies at INM, plasma membrane and endosomes." To make the Discussion an easier read, please explain these obvious parallels in a bit more detail. What was observed in these cited papers?

We have substantially revised this portion of the Discussion to clarify these points.

13) Discussion, fourth paragraph – please correct: accumulation of Chm7 in vps4d pom152d cells, not vps4d only.

Fixed.

14) Discussion, fifth paragraph: in the first line it is stated that the size of the necks at evaginations and herniations is suggestive of the presence of a spiraling polymer as observed in vitro and in vivo. To clarify this statement, it would be good to make explicit what McCullough et al. observed and with what proteins.

To include the additional requested discussion points, we ended up omitting an explicit discussion of the size of ESCRT-III spirals/tubes observed in many different in vivo and in vitro contexts.

15) Typo, "where the mechanism (of) Chm7 recruitment and activation can be decoupled."

Thank you, this is fixed.

16) How is ER distinguished from outer nuclear membrane herniations in tomograms? The authors imply ER membrane as a source involved in repair.

As the ER is contiguous with the outer nuclear membrane it is very difficult to differentiate between the two; moreover, because of this continuity if there is any membrane delivered for repair it is most logically from an ER source. Having said that, we completely agree that we don’t formally establish this and have weakened the language in the manuscript where this was implied.

17) If Chm7 is not required for the NE membrane remodeling (as seen in apq12- chm7-double deleted cells) how could the open-mutant active form of Chm7 on its own induce the membrane dynamics seen by tomography? In essence, Chm7 is not needed for this event to occur, but Chm7 induces it. Somehow, that logic needs to be fleshed out.

We now try to be more explicit about the differences between the herniations/INM evaginations that can be induced by Chm7 and those that are likely driven by NPC biogenesis. We also include a more detailed discussion to help elaborate on these points.

18) The accumulation of Chm7 at foci in Figure 1C would be more obvious with arrowheads noting a couple of these instances.

We tried several versions of this figure with arrowheads but ultimately left it as is because they made the images too busy and more difficult to interpret.

19) Why is the bottom row of images in Figure 1C displayed differently than the 2 upper rows?

This was necessary because there is no detectable cytosolic fluorescence in the bottom rows making it impossible to see cell boundaries. This is now explicitly stated in the figure legends.

20) The conclusion regarding the NLS of Lem2 in Kralt et al. is not as described in the manuscript. Kralt et al. found that the NLS of Lem2 is NOT like those of Heh1 or Heh2. Only that of Pom121 bears similarity.

Thank you for correcting this error.